# Solid-phase synthesis of protein-polymers on reversible immobilization supports

Hironobu Murata[1], Sheiliza Carmali[1,4], Stefanie L. Baker[1,2], Krzysztof Matyjaszewski[1,4] & Alan J. Russell[1,2,3,4,5]

Facile automated biomacromolecule synthesis is at the heart of blending synthetic and biologic worlds. Full access to abiotic/biotic synthetic diversity first occurred when chemistry was developed to grow nucleic acids and peptides from reversibly immobilized precursors. Protein–polymer conjugates, however, have always been synthesized in solution in multi-step, multi-day processes that couple innovative chemistry with challenging purification. Here we report the generation of protein–polymer hybrids synthesized by protein-ATRP on reversible immobilization supports (PARIS). We utilized modified agarose beads to covalently and reversibly couple to proteins in amino-specific reactions. We then modified reversibly immobilized proteins with protein-reactive ATRP initiators and, after ATRP, we released and analyzed the protein polymers. The activity and stability of PARIS-synthesized and solution-synthesized conjugates demonstrated that PARIS was an effective, rapid, and simple method to generate protein–polymer conjugates. Automation of PARIS significantly reduced synthesis/purification timelines, thereby opening a path to changing how to generate protein–polymer conjugates.

[1] Center for Polymer-Based Protein Engineering, Carnegie Mellon University, 5000 Forbes Avenue, Pittsburgh, PA 15213, USA. [2] Department of Biomedical Engineering, Scott Hall 4N201, Carnegie Mellon University, 5000 Forbes Avenue, Pittsburgh, PA 15213, USA. [3] Disruptive Health Technology Institute, Carnegie Mellon University, 5000 Forbes Avenue, Pittsburgh, PA 15213, USA. [4] Department of Chemistry, Carnegie Mellon University, 4400 Fifth Avenue, Pittsburgh, PA 15213, USA. [5] Department of Chemical Engineering, Carnegie Mellon University, 5000 Forbes Avenue, Pittsburgh, PA 15213, USA. Correspondence and requests for materials should be addressed to A.J.R. (email: alanrussell@cmu.edu)

Through pioneering work on PEGylated proteins[1,2], where a chain of polyethylene glycol (PEG) is coupled to a biomolecule, the merger of the synthetic and biologic worlds has saved countless lives and driven the application of biocatalysis in a variety of industries[1,3–7]. Protein PEGylation takes place in solution-based grafting-to syntheses where polymers are reacted with the protein surface[8,9]. This approach usually requires a large excess of polymer, is not easily controlled, and the density of modification can be limited by steric hindrance[10]. In the last decade, alternative routes to engineer the structure and function of proteins by growing polymers from their surfaces have been developed[11–15]. One common grafting-from approach uses atom-transfer radical polymerization (ATRP) from initiators that have been covalently attached to the surface of a protein. The high polymer grafting density and the potential for site-specific polymer growth that protein-ATRP achieves have enabled the synthesis of rationally designed functional protein–polymer conjugates[16] with dramatically enhanced stability[14,17,18] and therapeutic potential[19,20]. Growing polymers via ATRP from surface-initiated dissolved proteins is effective, but, the need to remove unreacted initiators, monomers, and catalysts in multiple purification steps has limited the automation of the process and its availability to a broad array of scientists. It can take weeks of careful synthesis and purification to generate just one protein–polymer variant. This challenge would be overcome by growing polymers from proteins that have been reversibly immobilized onto a solid surface.

The growth of reversibly immobilized peptides[21] and nucleic acids from solid supports has driven the emergence of automated syntheses and combinatorial chemistry. Perhaps more importantly, peptides and nucleic acid can be generated by non-expert biologists who interface with a simple device instead of a chemical reactor. Reversible immobilization of polymerization precursors on solid supports, such as polystyrene beads, was the foundation from which these elegant syntheses were built. We have been interested in how to reversibly immobilize an entire protein on a solid support, then subsequently react the immobilized protein with ATRP initiators (or other compounds of interest) before site-specific polymer growth. The resulting protein–polymer conjugate could then be released in a pure form from the solid support. Protein-ATRP on reversible immobilization supports (PARIS) would be a powerful transformer of the synthesis and impact of protein–polymer conjugates.

There are a variety of proven chemistries that can reversibly bind proteins to solid supports and some have been used to create grafted-to protein–polymer conjugates[22]. Non-covalent interactions[23] and hydrophobic adsorption[24] have been used, but the protein-support interactions are generally weak and protein-specific. Immobilized metal-affinity chromatography is widely used for purification of proteins containing an affinity tag, such as polyhistidine, but this is only applicable for recombinantly labeled proteins[25]. In addition, stable covalent disulfide bonds between free thiol groups on proteins and solid supports can be reduced to release bound protein[26]. In order to develop broadly applicable and predictable PARIS syntheses of protein–polymer conjugates, we searched for a covalent and reversible coupling chemistry that could be used with almost any protein. Solid supports functionalized with dialkyl maleic anhydrides can react with primary amine groups on all proteins. This reversible reaction is pH dependent with the complex dissociating at low pH[27]. The chemistry is highly suitable for proteins that are stable for brief periods at low pH (3–4), but can also be tailored for pH-sensitive proteins by increasing the reaction pH (5–6), albeit with lower efficiencies.

Herein, we explore immobilization of a protein's N-terminus α-amino and/or lysine ε-amino groups to dialkyl maleic anhydride-modified agarose beads, followed by ATRP from subsequently initiator-modified ε-amino groups on the protein surface, prior to protein–polymer conjugate release after reducing pH. PARIS-based synthesis of grafted-from protein–polymer conjugates opens the door to automated combinatorial syntheses and high throughput screening of next generation protein–polymer hybrids.

## Results

**PARIS chemistry.** Peptide synthesis from solid supports has traditionally used polystyrene resins[28]. Our initial experiments, however, demonstrated that non-specific hydrophobic adsorption of proteins to dialkyl maleic anhydride-modified polystyrene beads was significant. We therefore focused on hydrophilic supports that, we hypothesized, would reduce non-specific protein-support binding and ultimately be able to release a grown-from protein-polymer hybrid. Agarose beads are hydrophilic and are stable at extremes of pH, ionic strength, and in the presence of many denaturants. Currently, agarose beads are widely used in various chromatographic techniques for protein purification[29]. Dialkyl maleic anhydrides covalently react with primary amines above pH 6 and release below pH 6. Dialkyl maleic anhydride (DMA)-modified agarose beads (45–165 μm) were synthesized (Supplementary Fig. 1) and the pH dependence of on and off rates with a cyanine 3 amine fluorescent dye showed that up to $0.25\ \mu mol_{DM}\ mL^{-1}_{beads}$ would be available for protein attachment during PARIS. We next performed each of the four major steps for PARIS (Fig. 1): protein immobilization onto the DMA–agarose beads (pH 6.0 or 8.0); ATRP-initiator immobilization on the agarose-supported protein (using N-2-bromo-2-methylpropanoyl-β-alanine N′-oxysuccinimide bromide (NHS-Br) as previously described[17]); surface-initiated ATRP to grow polymers from agarose-supported proteins[15]; and, cleavage of the resulting protein–polymer conjugates from the agarose supports (below pH 6). Each step of PARIS was characterized with fourier-transform infrared spectroscopy (FT-IR) (Supplementary Fig. 2).

**Protein reaction with DMA–agarose.** Proteins contain a number of accessible amino groups, including the N-terminal α-amino and lysine side-chain ε-amino groups, that could potentially react with DMA–agarose beads to yield families of immobilized proteins. We hypothesized that we could preferentially target the protein–DMA reaction to the N-terminus of the protein by lowering the reaction pH, thereby generating mostly homogeneous protein–polymer conjugates. Previous studies have shown that acylation of α-amino groups (N-terminus) is preferred at pH 6.5 while ε-amino groups (lysine residues) react efficiently above pH 8.0[30]. Thus, we first investigated the pH dependence of DMA-lysine and DMA-N-terminal group reactions using Cy5.5 amine and glycyl–glycyl–Cy3 (GGCy3) fluorescent dyes (Supplementary Figs. 3–7) as lysine and N-terminal mimics, respectively. The p$K_a$ of a lysine side chain is approximately 10.5–12.0, while the p$K_a$ of the N-terminus is approximately 7.8–8.0. Thus, at a pH below 8.0, the N-terminus will have increased nucleophilicity over lysine residues, and we hypothesized that this would lead to preferential immobilization of the α-amino group to the DMA–agarose beads. Binding and release of these model dyes to the beads were determined as a function of pH and time (Supplementary Fig. 3). The data indicated that N-terminal α-amino-targeted protein binding to DMA beads was achievable at pH 6.0, providing evidence for site-specific immobilization (Supplementary Discussion).

Next, we investigated the pH dependence of protein immobilization at pH 6.0 and 8.0 and subsequent release over time from pH 3 to 6 using chymotrypsin (CT) as a model protein. At pH 6.0, both binding and release are occurring with variable rates

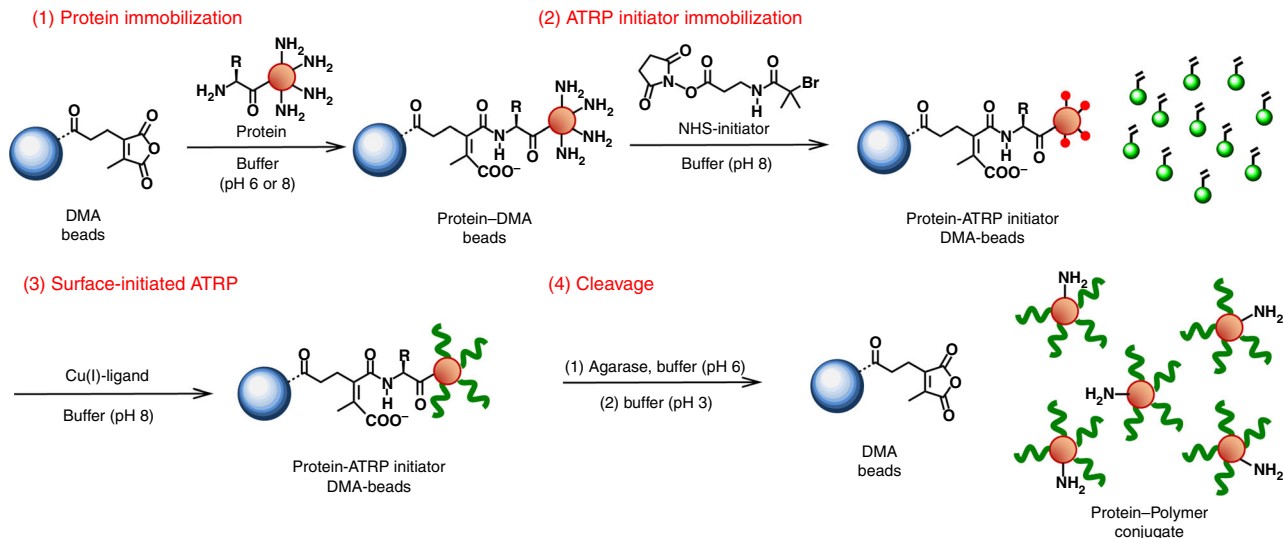

**Fig. 1** Solid-phase synthesis of protein–polymer conjugates by protein-atom transfer radical polymerization on reversible immobilization supports (PARIS). Protein is first immobilized onto DMA–agarose beads (at pH 6–8) through surface accessible primary amines. Remaining primary amines on the protein are then modified with ATRP initiators (NHS-Br), followed by surface-initiated ATRP, to create functional protein–polymer conjugates. The resulting protein–polymer conjugates are detached from the DMA–agarose beads below pH 6 and pass through a filter in pure form

depending on the reacting amino group. From the model dye data we knew that the binding rate at pH 6.0 was greater than the release rate for the *N*-terminal mimic, while the opposite was true for the lysine mimic (Supplementary Table 1). The total concentration of bound protein after immobilization (1.82 ± 0.12 and 4.02 ± 0.11 mg CT mL$^{-1}$ beads at pH 6.0 and 8.0, respectively) was determined using a bicinchoninic acid (BCA) assay. The concentration of immobilized protein achieved by reacting protein and DMA–agarose beads at pH 6.0 was less than that from the reaction at pH 8.0, again suggesting that this reaction could be site-specific since increased pH would increase the number of ε-amino groups that could react with the DMA, thereby significantly increasing the reaction stoichiometry.

Protein release kinetic studies were then performed over 60 min as a function of pH ranging (pH 3 to 6) for proteins initially immobilized at pH 6.0 and 8.0 (Fig. 2). In both cases, total protein recovered and release rate increased with decreasing pH consistent with the model dye experiments. The percent of protein recovered was calculated after 60 min for each releasing pH. The highest yields were 88 and 100% at pH 3 for initial immobilizations at pH 6.0 and 8.0, respectively, with the majority of release occurring within the first 5 min (Table 1). To ensure that CT was not inactivated by incubation in releasing buffer, residual activity was measured over time at pH 3 (Supplementary Fig. 8). CT was able to recover full activity at neutral pH even after incubation for 3 h at pH 3. For more sensitive proteins, however, release can be performed at pH 6.0 with 48% recovery. Moreover, in PARIS, the released product (a protein–polymer conjugate) may also have increased acid stability[31], further protecting the protein from the acidity of releasing buffer. Herein, we show that protein can be immobilized to and subsequently released from DMA–agarose beads with high recovery and maintained activity. We also show that release is pH dependent, albeit with varying degrees, allowing reaction conditions to be customized to match a given protein's sensitivity to acid. Importantly, the most likely application of PARIS will be to perform combinatorial syntheses of protein–polymer conjugates for subsequent high throughput screening. In this application, the final yield is less relevant than the ease and speed of synthesis. As long as we can generate enough protein polymer to assay, PARIS will have served its purpose.

Following immobilization, the next step in PARIS was ATRP-initiator (NHS-Br) modification of the remaining immobilized accessible protein amino groups. Our assumption was that after the protein was immobilized, the remaining amino groups would be available for ATRP-initiator modification. We further surmised from the model dye experiments that the ATRP initiator would not react with the *N*-terminus since it was already selectively bound to the beads in the case of pH 6.0 immobilization. CT has 15 primary amines: 14 lysine side chains and an additional α-amino group on the *N*-terminus. The number of initiator modifications for CT immobilized at pH 6.0 and 8.0 was determined to be 13 and 11, respectively, from matrix-assisted laser desorption/ionization time-of-flight mass spectrometry (MALDI-ToF MS) (Supplementary Fig. 9). We recently reported the use of MALDI-ToF MS after trypsin digestion of protein-initiator complexes to determine where initiators had reacted with proteins[13]. We used the same strategy to study modification sites on released initiator-modified CT. We did not observe modification of the *N*-terminal amino group by ATRP-initiator at pH 6.0, supporting our emerging view that DMA–agarose beads reacted with CT at the *N*-terminal amino group at low pH (Supplementary Fig. 10). In contrast, we observed *N*-terminal modification with ATRP initiator after immobilization at pH 8.0 indicated by the absence of the peptide fragment peak.

**PARIS synthesis of CT-conjugates.** Since the properties of CT-polymer conjugates have been studied in depth[32–38], we next focused on synthesizing and characterizing CT-polymer conjugates grown within, then released from, DMA–agarose beads. For the polymer, we decided to grow poly(carboxybetaine methacrylate) (pCBMA), a hydrophilic and zwitterionic polymer, from the surface of the initiated and reversibly immobilized enzyme. Zwitterionic polymers stabilize CT against irreversible inactivation at extremes of temperatures and pH [17,39].

We expected that CT-pCBMA conjugates would grow from initiator sites on protein–DMA–agarose beads. We were particularly interested in whether the pH of immobilization and the agarose beads themselves would impact the structure and function of the subsequently released enzyme. The chemical structure of CT-pCBMA was initially characterized with $^1$H NMR and FT-IR (Supplementary Figs. 11, 12). We further compared

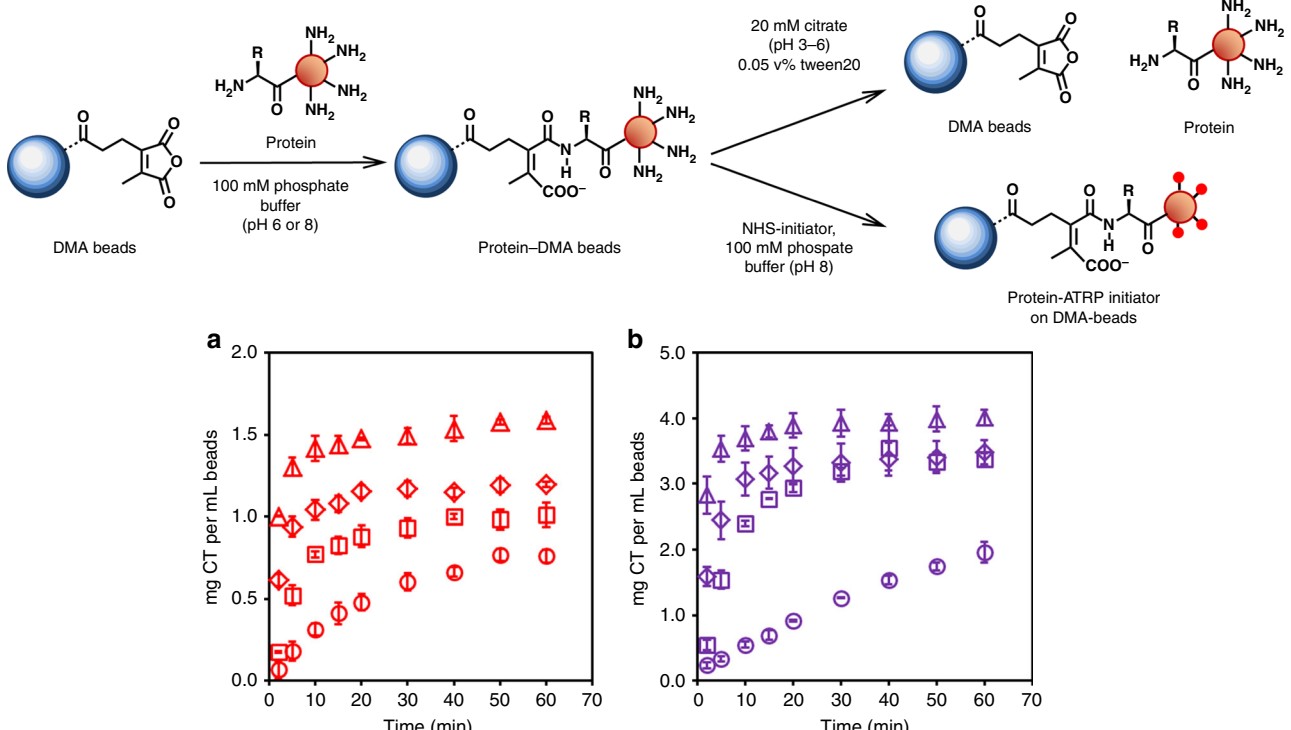

**Fig. 2** Reversible protein immobilization to DMA–agarose beads and ATRP initiator modification. **a** Chymotrypsin release kinetics as a function of pH (3–6) after initial immobilization at pH 6.0 (red, determined by microBCA protein assay) and **b** after initial immobilization at pH 8.0 (purple, determined by microBCA assay). In both plots, release was performed by incubation at pH 3 (open triangle), pH 4 (open diamond), pH 5 (open square), and pH 6 (open circle). The amount of recoverable protein and release rate increased as pH decreased. Error bars represent standard deviation from triplicate measurements

the released PARIS-CT conjugates to both solution-based CT-pCBMA conjugates and native CT (Fig. 3). Following release from the DMA–agarose beads, protein concentration was determined spectroscopically. We first characterized released CT-pCBMA conjugate hydrodynamic diameter ($D_h$) using dynamic light scattering (DLS). The released conjugates grew in $D_h$ from 4.4 nm (native CT) to approximately 9 nm, independent of immobilization pH. Solution-based CT-pCBMA conjugates were slightly larger than PARIS-synthesized conjugates (Table 2). We further characterized the conjugates through acid hydrolysis to cleave the polymer followed by gel permeation chromatography (GPC) showing polymer molecular weights of 9.2 and 8.2 kDa for CT immobilized at pH 6.0 and 8.0, respectively with low polydispersity indices (PDI) for each.

We further compared the activity of released PARIS-CT conjugates to native CT and CT-pCBMA grown in solution. The turnover number and Michaelis constant ($k_{cat}$ and $K_M$) showed that all conjugates had similar activities with *N*-succinyl-L-Ala-L-Ala-L-Pro-L-Phe-*p*-nitroanilide (suc-AAPF-pNA) (Table 2). Since the PARIS conjugates immobilized at pH 6.0 and 8.0 had similar activities, the location of protein-bead immobilization did not significantly alter CT activity. Naturally, if polymer growth from the terminal amino group was performed on a protein that was sensitive to *N*-terminal modification, we would expect the activity of the conjugate to be lower from PARIS conjugates immobilized at pH 6.0. It is also worth noting that protein–polymer conjugates typically have reduced catalytic efficiencies through a combined decrease in $k_{cat}$ due to structural stiffening[40] and decrease in $K_M$ due to the polymer's super-hydrophilicity[17]. The overall catalytic productivity, $k_{cat}/K_M$, of PARIS CT-pCBMA was similar to both solution-synthesized CT-pCBMA and native CT.

**Table 1 Protein recovered in releasing buffer from pH 3–6**

| Releasing pH | Percent recovery from pH 6.0 immobilization | Percent recovery from pH 8.0 immobilization |
| --- | --- | --- |
| pH 3 | 87.8 ± 5.9 | 99.8 ± 4.1 |
| pH 4 | 68.8 ± 4.4 | 86.5 ± 5.3 |
| pH 5 | 55.6 ± 5.5 | 84.1 ± 3.1 |
| pH 6 | 41.8 ± 3.5 | 47.8 ± 4.1 |

The release rate and total amount of protein recovered are pH dependent with the highest recovery at pH 3 after 60 min. Error bars indicate standard deviation from triplicate measurements
Errors indicate standard deviation from triplicate measurements

Next, we sought to demonstrate that PARIS conjugates maintained the same stabilizing effect as conventionally synthesized protein–polymer conjugates in solution. Protein–polymer conjugates synthesized in solution have shown enhanced stability to extremes of pH[41], temperatures[18], and organic solvents[42] due to the protective polymer coating. PARIS-CT conjugates significantly enhanced the thermostability of the enzyme at 50 °C (Fig. 3). Native CT was irreversibly inactivated after approximately 2 h at 50 °C. Both PARIS CT-pCBMA and solution-based CT-pCBMA had significantly increased stability. Interestingly, PARIS conjugates that were initially immobilized at pH 6.0 were significantly more stable after 6 h at 50 °C than those from pH 8.0. Previous studies have shown that many proteins have regions that are particularly susceptible to stability-impacting modifications and in many proteins the *N*-terminus is the most thermally sensitive region[43–46]. Additionally, enzymes with increased thermostability are often more hydrophobic[47,48]. Since the *N*-terminus in CT was located in a largely hydrophobic

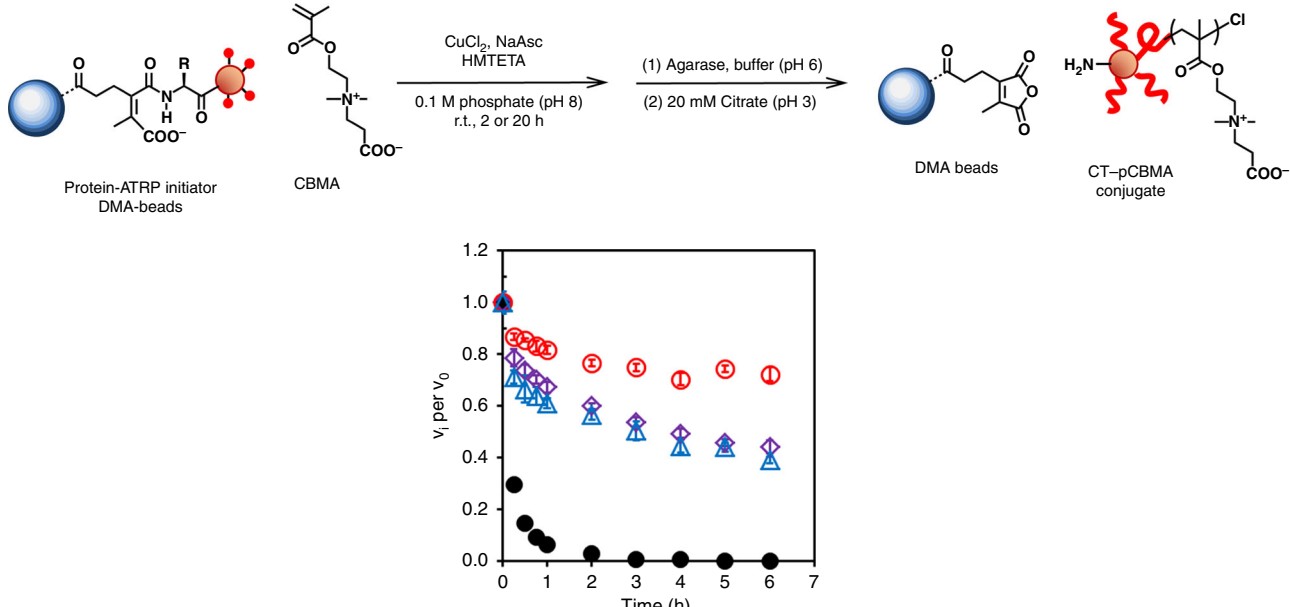

**Fig. 3** Properties of released PARIS-synthesized protein–polymer conjugates. Irreversible inactivation of native CT and CT−pCBMA at 50 °C. CT−pCBMA immobilized at pH 6.0 and polymerized for 2 h (red open circle), CT−pCBMA immobilized at pH 8.0 and polymerized for 2 h (purple open diamond), CT−pCBMA by solution synthesis (blue open triangle), and native CT (black closed circle). Samples were incubated in 100 mM sodium phosphate buffer (pH 8.0, 50 °C) at 3.9 µM CT and residual activity was measured over 6 h. All CT-pCBMA conjugates showed enhanced thermostability compared to native CT. CT-pCBMA immobilized at pH 6.0 with a free N-terminus showed the highest thermostability. Error bars represent standard deviation from triplicate measurements

### Table 2 Characterization of released PARIS-synthesized CT conjugates

| CT immobilization (pH) | Released CT-pCBMA[b] (mg CT per mL beads) | $D_h{}^c$ (nm) | Cleaved polymer[d] $M_n$ (kDa); ($M_w/M_n$) | Estimated conjugate $M_w{}^e$ (kDa) | $k_{cat}{}^f$ (s$^{-1}$) | $K_M{}^f$ (µM) | $k_{cat}/K_M{}^f$ (s$^{-1}$ µM$^{-1}$) |
|---|---|---|---|---|---|---|---|
| 6 | 0.56 ± 0.02 | 9.2 ± 2.4 | 9.2;(1.27) | 115.7 | 33.3 ± 1.0 | 65.5 ± 8.2 | 0.508 ± 0.065 |
| 8 | 1.23 ± 0.01 | 9.1 ± 1.9 | 8.2;(1.26) | 133.1 | 34.6 ± 1.2 | 70.4 ± 10.5 | 0.491 ± 0.075 |
| Native CT | – | 4.4 ± 1.3 | – | – | 34.6 ± 1.4 | 80.3 ± 13.1 | 0.431 ± 0.072 |
| Solution-based CT-pCBMA[a] | – | 10.9 ± 1.4 | – | – | 21.5 ± 0.8 | 45.0 ± 7.9 | 0.478 ± 0.086 |

[a] CT-pCBMA conjugate was prepared by solution-based method
[b] The concentration of released conjugate based on CT per 1 mL of beads (estimated by UV absorption assay) indicating that there are more possible binding sites at pH 8.0 than at pH 6.0
[c] Hydrodynamic diameters (number intensity) of the native CT and CT-pCBMA conjugates were measured using dynamic light scattering in 20 mM sodium citrate (pH 3.0) at 25 °C showed an increase in conjugate size over native CT
[d] Number average molar mass of cleaved pCBMA and polydispersity index from GPC
[e] Estimated conjugate molecular weight from GPC data
[f] Michaelis–Menten kinetic values for CT-catalyzed hydrolysis of suc-AAPF-pNA were determined by nonlinear curve-fitting of plots of initial rate versus substrate concentration using Enzfitter software. Conjugates synthesized by PARIS did not alter activity in comparison to solution-synthesized CT-pCBMA and native CT

area, disruption of this hydrophobicity by initiator modification and consequent hydrophilic polymer growth could have decreased the thermostability of the pH 8.0 PARIS CT-pCBMA.

**ATRP polymerization kinetics**. We next explored whether DMA–agarose beads used in PARIS disrupted ATRP reaction kinetics. Thus, ATRP kinetics of PARIS and solution-synthesized conjugates were compared (Tables 3–4). Polymer growth was monitored over 60 min for a fixed monomer concentration of 25 mM by measuring $D_h$ of conjugates at specified time points using DLS (Table 3 and Supplementary Fig. 13). Additionally, polymers were cleaved from the conjugates at each specified time point, and their molecular weights and PDIs were measured using GPC (Supplementary Fig. 14). Polymer molecular weight did not increase significantly after 5 min indicating that the ATRP reaction was fast and completed within the first 5 min. Comparison of PARIS and solution-based conjugates showed similar overall

polymer growth, however, PARIS conjugates displayed lower PDIs. The agarose bead pore size used in PARIS was approximately 30 nm, which could have provided a uniform microenvironment during chain propagation leading to lower PDIs and more uniform conjugates. In conventional solution-based ATRP, polymer molecular weight can be tuned to a desired value by increasing the monomer to initiator ratio during the ATRP reaction. To show that conjugate size could be easily varied by PARIS, the monomer concentration was systematically increased from 12.5 to 100 mM and conjugate $D_h$, polymer molecular weights, and PDIs were determined for a 60 min reaction time (Table 4). As expected, conjugate $D_h$ and polymer molecular weight increased with increasing monomer concentration for both PARIS and solution-based conjugate synthesis (each maintaining low PDIs) (Supplementary Figs. 13, 14). In summary, we were excited to observe that for pCBMA, highly uniform conjugates were synthesized by PARIS within 5 min, polymer

**Table 3 PARIS and solution-based conjugate ATRP kinetics**

| Sample[a] | $D_h$[b] | Cleaved pCBMA[c] | | |
|---|---|---|---|---|
| | (nm) | $M_n$ | $M_w$ | PDI |
| Solution | | | | |
| 5 min | 12.3 ± 4.2 | 10100 | 14400 | 1.42 |
| 10 min | 13.5 ± 3.9 | 9800 | 13900 | 1.44 |
| 20 min | 12.8 ± 7.5 | 10000 | 14700 | 1.47 |
| 40 min | 11.1 ± 7.3 | 9800 | 14200 | 1.45 |
| 60 min | 11.3 ± 7.8 | 11000 | 16700 | 1.50 |
| PARIS | | | | |
| 5 min | 7.3 ± 1.2 | 9600 | 12300 | 1.28 |
| 10 min | 7.9 ± 2.5 | 9900 | 12800 | 1.29 |
| 20 min | 9.3 ± 1.3 | 9600 | 12500 | 1.30 |
| 40 min | 8.6 ± 2.0 | 10000 | 13300 | 1.33 |
| 60 min | 9.4 ± 1.4 | 10600 | 14700 | 1.39 |

Fixed monomer concentration of 25 mM. Chain propagation was completed within 5 min for CT-pCBMA for both PARIS and solution-based approaches. Polymers synthesized by PARIS also had lower dispersities than solution-based

**Table 4 PARIS and solution-based conjugate ATRP kinetics**

| Sample[a] | $D_h$[b] | Cleaved pCBMA[c] | | |
|---|---|---|---|---|
| | (nm) | $M_n$ | $M_w$ | PDI |
| Solution | | | | |
| 12.5 mM | 9.0 ± 2.9 | 8200 | 9900 | 1.18 |
| 25 mM | 11.8 ± 2.4 | 11100 | 15100 | 1.35 |
| 50 mM | 13.6 ± 4.3 | 16100 | 25600 | 1.61 |
| 100 mM | 17.0 ± 8.4 | 27900 | 48800 | 1.77 |
| PARIS | | | | |
| 12.5 mM | 7.4 ± 3.9 | 7000 | 7700 | 1.10 |
| 25 mM | 9.4 ± 2.5 | 9600 | 12500 | 1.30 |
| 50 mM | 13.5 ± 3.2 | 15200 | 21500 | 1.43 |
| 100 mM | 15.2 ± 2.3 | 23400 | 36000 | 1.53 |

Increasing monomer concentration for a 60-min reaction time. Polymer molecular weight, and thus conjugate size, can be predictably increased by increasing monomer concentration

molecular weights were predictably controlled, and PARIS conjugates were similar, perhaps even more homogeneous, compared to solution-synthesized conjugates.

**PARIS synthesis of protein conjugates**. Although our results with CT conjugates synthesized by PARIS were exciting, an important step was to demonstrate that PARIS could be applied to a breadth of unrelated proteins. We sought to demonstrate that solution-synthesized conjugates were nearly identical to PARIS-synthesized conjugates. To do this, we selected a series of proteins with varying sizes, structures, and N-terminii accessibility: lysozyme (Lyz, $M_{w,monomer} = 14.3$ kDa), avidin ($M_{w,tetramer} = 68$ kDa), acetylcholinesterase (AChE, $M_{w,tetramer} = 272$ kDa), and uricase (Uox, $M_{w,tetramer} = 140$ kDa). As with CT, we first determined whether the reaction between DMA–agarose and protein was N-terminus specific for each protein. Our combined data from all proteins showed that the immobilization reaction was N-terminus selective at pH 6.0, as long as the α-amino group was surface accessible, and was independent of protein size and quaternary structure (Supplementary Figs. 15, 18, Supplementary Table 2, and Supplementary Discussion). Initiator-modified proteins were also characterized by MALDI-ToF MS to determine the number of modification sites after initial immobilization at pH 6.0 and 8.0 (Supplementary Figs. 19–21). After successful protein-ATRP with pCBMA, followed by release, the protein–polymer conjugates were fully characterized (Table 5). The percent of recovered conjugate was decreased in comparison to the prior released native protein experiments, with the highest conjugate recovery obtained from the smallest starting protein size (51% for lysozyme). This result was not surprising since larger conjugates could become more trapped inside the pores after polymer growth, thus hindering full release. We are currently exploring a number of methods to optimize release. For example, agarose bead pore size can be increased to accommodate larger proteins, multiple incubation steps in releasing buffer can be performed in series, or the ratio of releasing buffer to agarose bead solution can be increased. We have also been encouraged by early results with agarase-induced release optimization.

Hydrodynamic diameters were also similar for each PARIS-synthesized and solution-synthesized conjugate pair. Polymers were also cleaved and analyzed using GPC to provide complete characterization. All polymers maintained low PDIs whether synthesized by PARIS or in solution.

We also were interested in whether each of the PARIS-synthesized conjugates would have the same activity as solution-synthesized conjugates. We note here that in this paper we were not

seeking to optimize conjugate activity for each of the proteins used. Activity assays were performed, specific for each enzyme (Supplementary Tables 3–6), and activities were reported as a ratio of PARIS-synthesized to solution-synthesized conjugates (Table 5e). In all cases, PARIS conjugates had maintained activities compared to solution-based conjugates. Additionally, lysozyme and uricase conjugates synthesized by PARIS had twice the activity over their solution-based counterparts. We have not performed extensive enough experiments in order to be able to claim that PARIS generates more active conjugates, but the data are promising. For acetylcholinesterase, we discovered that the enzyme was sensitive to release at pH 3, but successful release at pH 5 enabled a comparison with solution-synthesized conjugate (Supplementary Fig. 22). Overall, we were pleased to observe that conjugate synthesis via PARIS chemistry was not only suitable for a breadth of proteins, but also easily tunable for a specific protein's sensitivity. While we have not yet focused on yield optimization, the PARIS strategy is not limited to the release chemistry we selected in this initial demonstration of solid-state protein-ATRP. The PARIS data were comparable to conventional solution-generated data in terms of both physical and functional bioconjugate properties. PARIS is a reliable protein–polymer conjugate synthesis method that would be straightforward for any biologist to perform.

Facile automated one-pot PARIS. We developed PARIS in order to be able to generate protein–polymer conjugates in hours versus weeks, and to simplify and automate the chemistry involved. The conceptual attractiveness of flowing reactants into a column reactor, removing unreacted initiator, removing unreacted monomers, and purifying the conjugates by release from the beads drove us to design a PARIS-flow reactor (Fig. 4a). CT was bound to the DMA–agarose beads, reacted with initiator, and then released from the DMA beads in the reactor after polymerization. PARIS synthesis of CT-pCBMA in the flow reactor was compared to batch synthesis. Our data showed that conjugates synthesized in batch-mode and flow-mode had similar structure and function. The $D_h$ of the conjugates released from the automated flow-based reactor were 8.7 and 9.2 nm for CT initially immobilized at pH 6.0 and 8.0, respectively (Table 6). Similarly (and in agreement with results from the batch synthesis), both PARIS conjugates synthesized in the flow reactor had enhanced thermostability at 50 °C relative to native CT, while the most thermostable conjugate was CT-pCBMA that had been immobilized at pH 6.0.

Conventional protein–polymer conjugates that are grown from proteins require initiator modification, followed by days of separation and purification of initiator-modified proteins from

**Table 5 PARIS synthesis and characterization of ranging proteins**

| Protein | Protein on beads[a] | Released protein–pCBMA[b] | $D_h$[c] | Cleaved Polymer[d] | Ratio of activity[e] |
|---|---|---|---|---|---|
| | (mg protein per mL beads) | (mg protein per mL beads) (% recovery) | (nm) (solution) | $M_n$ (kDa) ($M_w/M_n$) (solution) | (PARIS: solution) |
| Lysozyme | 1.89 | 0.97 (51%) | 9.4 ± 1.8 (7.1 ± 2.6) | 19.8 (1.37) (15.0 (1.26)) | 1.94 ± 0.13 |
| Avidin | 3.54 | 1.05 (30%) | 13.2 ± 2.8 (21.7 ± 6.4) | 13.2 (1.30) (16.1 (1.34)) | 1.01 ± 0.14 |
| Chymotrypsin | 2.01 | 0.89 (44%) | 8.1 ± 0.7 (10.9 ± 1.4) | 10.6 (1.39) (11.0 (1.50)) | 1.03 ± 0.24 |
| Acetylcholinesterase | 0.79 | 0.16 (20%) | 13.0 ± 2.0 (13.9 ± 5.2) | 14.5 (1.35) (7.2 (1.33)) | 1.09 ± 0.08 |
| Uricase | 0.43 | 0.09 (21%) | 10.9 ± 1.7 (12.5 ± 5.0) | 32.4 (1.50) (14.7 (1.30)) | 2.33 ± 0.15 |

[a] Concentration of immobilized protein per 1 mL of beads determined by microBCA protein assay
[b] Concentration of released conjugate per 1 mL of beads and percentage of recovered protein determined by microBCA protein assay
[c] Hydrodynamic diameter measured by dynamic light scattering (number distribution)
[d] Number average molecular weight and polydispersity index of cleaved pCBMA from PARIS conjugates determined by gel permeation chromatography and compared to solution-based conjugates
[e] Ratio of conjugate activity of PARIS to solution-based approaches. Errors represent standard deviation from triplicate measurements

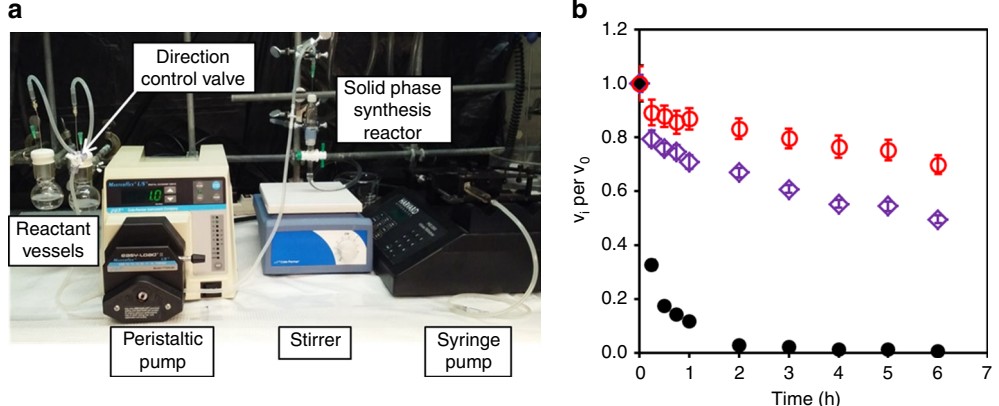

**Fig. 4** Automated one-pot rapid PARIS protein–polymer conjugate synthesis and properties. **a** The experimental setup of a flow reactor for PARIS. **b** Irreversible inactivation of native CT and CT-pCBMA at 50 °C. CT-pCBMA synthesized in the flow reactor immobilized at pH 6.0 (red open circle) or pH 8.0 (purple open diamond) and native CT (black closed circle) were incubated in 100 mM sodium phosphate buffer (pH 8.0, 50 °C) at 3.9 µM CT for 6 h. Both CT-pCBMA conjugates showed enhanced thermostability compared to native CT and CT-pCBMA immobilized at pH 6.0 with a free N-terminus showed the highest thermostability similar to batch mode studies. Error bars represent standard deviation from triplicate measurements

**Table 6 Characterization of CT conjugates by PARIS flow reactor**

| CT immobilization | Protein on beads[a] | Released CT-initiator[b] | Number of initiators on the conjugate[c] | Released CT-pCBMA by UV abs[d] | Released CT-pCBMA by activity[e] | $D_h$[f] |
|---|---|---|---|---|---|---|
| (pH) | (mg CT per mL beads) | (mg CT per mL beads) | | (mg CT per mL beads) | (mg CT per mL beads) | (nm) |
| 6 | 1.1 | 0.6 | 7.2 | 0.37 | 0.13 | 8.7 ± 0.1 |
| 8 | 2.2 | 1.7 | 10.4 | 1.53 | 0.96 | 9.2 ± 0.4 |

[a] Concentration of immobilized CT determined by a microBCA protein assay. The concentration of CT on the beads at pH 6.0 is less than pH 8.0 indicating more possible binding sites at pH 8.0
[b] Concentration of released CT-initiator determined by a microBCA protein assay
[c] Average number of initiators per CT determined by fluorescamine amine assays using standard protocols
[d] Concentration of released conjugates estimated by UV absorption
[e] Concentration of released conjugates estimated by enzyme activity using suc-AAPF-pNA as a substrate
[f] Hydrodynamic diameter of the CT-pCBMA conjugates was measured using dynamic light scattering in 20 mM sodium citrate (pH 3.0) at 25 °C showing an increase in conjugate size over native CT ($D_h$ = 4.4 nm, number distribution)

excess initiator, and then 2–4 days of polymerization and final purifications. The flow reactor experiment described above was completed in less than 6 h. While the individual steps of conjugate synthesis between solution-based and PARIS-based approaches were similar, the PARIS purification steps were rapid and readily automated. This reduction in synthesis and purification time, coupled with our ability to multiplex the system, removed the complexity barrier from protein–polymer conjugate

synthesis. It is currently difficult to predict how polymer conjugation will affect the resulting conjugate's overall function. Since PARIS conjugates were similar in structure and function to solution-based conjugates, an attractive feature of the flow reactor design was that high throughput synthesis would allow more rapid screening of a multitude of protein–polymer conjugates.

PARIS is a synthetic approach that allows solid-phase synthesis of grafted-from protein–polymer conjugates. PARIS generates

conjugates with similar structure and function to traditional protein-ATRP in solution. Importantly, PARIS can be performed in a simple flow reactor, opening the door to automated and combinatorial protein–polymer conjugate syntheses. We believe that PARIS will have a significant impact on the accessibility of protein–polymer conjugate synthesis to a broad array of protein scientists and engineers.

## Methods

The experiments were not randomized and the investigators were not blinded to allocation during experiments and outcome assessment. If not stated otherwise, measurements were performed in triplicate and error bars represent standard deviation.

**Materials**. α-CT from bovine pancreas (type II), Lysozyme from chicken egg white, Acetylcholinesterase from *Electrophorus electricus* (electric eel, type VI-S), Uricase from porcine liver (type V), and Agarase from *Pseudomonas atlantica* were purchased from Sigma Aldrich (St Louis, MO). Avidin from *Gallus gallus* egg white was purchased from Lee biosolutions (Maryland Heights, MO). Protein surface active ATRP initiator (NHS-Br) was prepared as described previously [15].

**Preparation of dialkyl maleic anhydride (DMA) agarose beads**. All materials were purchased from Sigma Aldrich (St Louis, MO) and used without further purification unless stated otherwise. Aminated agarose beads (ω-Aminohexyl–Sepharose® 4B, 10 mL, swollen, 7–12 μmol $NH_2$ mL$^{-1}$ beads) were washed with deionized water (30 mL × 2), 20 mM citric acid (30 mL × 2) and 100 mM sodium phosphate buffer (pH 9, 30 mL × 2). A pre-incubated solution of 2,5-dihydro-4-methyl-2,5-dioxo-3-furanpropanoic acid (44 mg, 240 μmol, TCI America, Philadelphia, PA), EDC·HCl (46 mg, 240 μmol) and 1-hydroxybenzotriazole hydrate (32 mg, 240 μmol) in dimethylformamide at 0 °C for 30 min was added to the aminated agarose solution with triethylamine (70 μL, 500 μmol), and the mixture was shaken at room temperature for 30 min. The beads were washed with deionized water (30 mL × 3), 20 mM citric acid (30 mL × 3) and deionized water (30 mL × 2). To block unmodified amine groups, the beads were incubated with acetic anhydride (71 μL, 500 μmol) and triethylamine (70 μL, 500 μmol) in deionized water (30 mL) at room temperature for 30 min. The beads were then washed as previously described then stored in the refrigerator.

**Quantitative analysis of accessible DM group on beads**. Prepared DMA beads (10 μL) were placed in a solution of 40 μM Cyanine3 amine (Lumiprobe, Hallandale Beach, FL) in 100 mM sodium phosphate buffer (pH 8) containing 0.05 v/v% Tween 20 (500 μL) and shaken at room temperature for 60 min. Beads were settled by centrifugation and the supernatant was aspirated. The beads were then washed with 100 mM sodium phosphate (pH 8) containing 0.05 v/v% Tween 20 (1 mL × 5). After removing the supernatant, 20 mM sodium citrate (pH 3) containing 0.05 v/v% Tween 20 (1 mL) was added to the beads and shaken at room temperature for 60 min. Supernatant fluorescence intensities from the releasing solution were measured at an excitation of 550 nm and an emission of 570 nm with 10 nm bandwidths by a Safire2 plate reader (Tecan, Group Ltd.). Concentrations were calculated from standard curves.

**Preparation of glycyl–glycyl–Cy3 (GGCy3)**. *N,N′*-Diisopropylcarbodiimide (770 μL, 5.0 mmol) was added to a solution of Boc–GlyGly–OH (Bachem America, Torrance, CA, 920 mg, 4.0 mmol) and *N*-hydroxysuccinimide (NHS, 575 mg, 5.0 mmol) in dichloromethane (50 mL) at 0 °C. The solution was stirred at room temperature overnight. Precipitated urea was filtered out and the filtrate was evaporated to remove dichloromethane under vacuum. Boc–GlyGly–NHS was isolated by recrystallization in 2-propanol with a 63% yield verified with proton nuclear magnetic resonance spectroscopy (Supplementary Figs. 5–7) recorded in CDCl₃ using a 300 MHz, Bruker Avance in the NMR facility located in Center for Molecular Analysis, Carnegie Mellon University, Pittsburgh, USA. The solution of Boc–GlyGly–NHS (16 mg, 48 μmol) in chloroform (40 mL) was added to a solution of Cyanine3 amine (25 mg, 40 μmol) and triethylamine (7 μL, 50 μmol) in dimethyl sulfoxide (DMSO; 100 μL) and stirred at room temperature overnight. The mixture was washed with 0.5 N hydrochloric acid (HCl) aq. (50 mL × 2), saturated NaHCO₃ (50 mL × 2), and saturated NaCl (50 mL × 2), then dried with MgSO₄. After MgSO₄ removal by filtration, Boc–GlyGly–Cy3 was isolated by evaporating chloroform under vacuum with an 89% yield verified by $^1$H NMR. The mixture of Boc–GlyGly–Cy3 (30 mg, 35.6 μmol) in 4 M HCl, 1,4-dioxane (80 μL), and 1,4-dioxane (920 μL) was stirred at room temperature for 4 h. GlyGlyCy3 was obtained by evaporation of dioxane under vacuum with an 98% yield verified with $^1$H NMR.

GGCy3, oily compound, $^1$H NMR (300 MHz, CDCl₃) δ 1.4–1.9 (broad, 14H, 7 × CH₂ and 12H, Cy3–CH₃), 2.5 (broad, 2H, CH₂), 3.2 (broad, 4H, 2 × CH₂), 3.8 (broad, 3H, Cy3 N–CH₃), 4.0–4.3 (broad, 2H, CH₂ and 4H, 2 × Gly$^\alpha$), 6.8 (broad, 2H, CH=CH), 7.3 and 7.4 (broad, 8H Cy3–Ar H), 8.2 (broad, 1H, amide), 8.4 (broad, 1H, CH=CH–CH), 8.6 (broad, 2H, amide) ppm; $^{13}$C NMR (75 MHz, DMSO-d₆) δ 25.2, 25.7, 26.1, 26.8, 28.0, 28.2, 28.7, 28.9, 29.0, 29.1, 29.7, 34.0, 35.0,

36.1, 38.7, 39.7, 42.9, 49.0, 49.2, 103.9, 105.6, 116.0, 125.7, 127.5, 128.5, 131.2, 140.5, 141.9, 169.3, 170.1, 173.8 ppm; IR (NaCl plate) 2956, 2924, 2853, 1712, 1651, 1557, 1493, 1456, 1416 and 1376 cm$^{-1}$; HRMS (*m/z*): [M − 2H]$^+$ calcd. for $C_{40}H_{57}N_6O_3^{2+}$, 670.46; found, 670.94.

**GGCy3 and Cy5.5 amine binding to DMA beads**. Five hundred microliters of GGCy3 or Cy5.5 amine solution (Lumiprobe, Hallandale Beach, FL, 40 μM in 100 mM sodium phosphate (pH 5 – 8) containing 0.05 v/v% Tween 20 and 20 μL of DMA beads were shaken at room temperature. The supernatant was removed at given incubation times and the beads were washed with incubation buffer (1 mL × 2) followed by washing buffer (100 mM sodium phosphate (pH 8) containing 0.05 v/v% of Tween 20 (1 mL × 3) to remove residuals. The beads were incubated in releasing buffer (1 mL, 20 mM sodium citrate (pH 3) containing 0.05 v/v% Tween 20) at room temperature for 1 h, and supernatant fluorescence intensities were measured with a Safire2 plate reader (GGCy3: excite 550 nm, emit 570 nm; Cy5.5: excite 670 nm, emit 707 nm). Concentrations were calculated from standard curves. See Supplementary Methods.

**GGCy3 and Cy5.5 amine release from DMA beads**. Thirty microliters of DMA beads were pre-incubated with 750 μL of GGCy3 or Cy5.5 amine solution (40 μM in 100 mM sodium phosphate (pH 8) containing 0.05 v/v% Tween 20). The supernatant was removed and the dye bound beads were washed with the washing buffer. The beads were incubated in releasing buffer (1.5 mL) at room temperature and fluorescence intensities of supernatant aliquots (100 μL) at given time point were measured at wavelengths mentioned above. See Supplementary Methods.

**Protein immobilization on DMA beads**. A volume of 4.5 mL of protein solution (2 mg mL$^{-1}$, 100 mM sodium phosphate (pH 6 or 8) containing 0.05 v/v% Tween 20 was combined with 1.5 mL of DMA beads in the solid phase peptide synthesis vessel (10 mL capacity, Chemglass) and shaken at room temperature or in refrigerator for 30 min. After removing the supernatant, the beads were washed with incubation and washing buffers. The amount of immobilized protein on the DMA beads was determined using a Micro BCA Protein Assay Kit (ThermoFisher Scientific). The sample solution and beads (10 μL) in 500 μL of deionized water were mixed with micro BCA working reagent (500 μL) and incubated at 60 °C for 1 h. The absorbance at 562 nm was recorded by a UV–VIS spectrometer (Lambda 2, Perkin Elmer).

**ATRP initiator modification onto immobilized protein**. Two hundred millimolar of ATRP initiator, NHS-Br[15] (NHS-Br) solution in DMSO (225 μL) was added to the suspension of the protein immobilized DMA beads (1.5 mL) in 100 mM sodium phosphate (pH 8, 4.5 mL) and shaken at room temperature for 30 min. The beads were washed with 100 mM sodium phosphate (pH 8, 5 mL × 5). Estimation of immobilized ATRP initiator on protein was carried out by BCA and fluorescamine assays. The beads (20 μL) were incubated in 20 mM sodium citrate (pH 3) containing 0.05 v/v% of Tween 20 (200 μL) at room temperature for 1 h. BCA protein assay was used to determine protein concentration in the supernatant as previously described. Fluorescamine assay was used to determine the number of bound initiators. Aliquots (40 μL) of supernatant, 100 mM sodium phosphate (40 μL, pH 9), and fluorescamine solution in DMSO (20 μL, 3 mg mL$^{-1}$) were added into a 96-well plate and incubated at room temperature for 15 min. Fluorescence intensities were measured at the excitation of 390 nm and emission of 470 nm with 10-nm bandwidths by a Safire2 plate reader. Concentration was determined using a standard curve.

**Trypsin digestion of protein-initiator conjugates**. Trypsin digests were used to generate peptide fragments from which protein-initiator attachment sites could be determined using matrix-assisted laser desorption/ionization time-of-flight (MALDI-ToF) mass spectrometry. Five proteins were studied. α-CT from bovine pancreas (type II), lysozyme from chicken egg white, acetylcholinesterase from *Electrophorus electricus* (electric eel, type VI-S), uricase from porcine liver (type V). Avidin from *Gallus gallus* egg white was purchased from Lee biosolutions (Maryland Heights, MO). Samples were digested according to the protocol described in the In-Solution Tryptic Digestion and Guanidination Kit. Briefly, 10–20 μg of protein or protein-initiator complexes (10 μL of a 2 mg mL$^{-1}$ protein solution in deionized water) were added to 15 μL of 50 mM ammonium bicarbonate with 1.5 μL of 100 mM dithiothreitol in a 0.5 mL Eppendorf tube. The reaction was incubated for 5 min at 95 °C. Thiol alkylation was conducted by the addition of 3 μL of 100 mM iodoacetamide aqueous solution to the protein solution and incubation in the dark for 20 min at room temperature. Following incubation, 1 μL of 100 ng trypsin was added to the protein solution and the reaction was incubated at 37 °C for 3 h. Then, an additional 1 μL of 100 ng trypsin was subsequently added. The reaction was terminated after a total reaction time of 5 h by the addition of trifluoroacetic acid (TFA). Digested samples were purified using ZipTipC₁₈ microtips and eluted with 2 μL of matrix solution (20 mg mL$^{-1}$ sinapinic acid in 50% acetonitrile with 0.1% TFA) directly onto a MALDI-ToF plate for subsequent analysis. The molecular weight of the expected peptide fragments before and after digestion was predicted using PeptideCutter (ExPASy Bioinformatics Portal, Swiss Institute of Bioinformatics). Peptide fragment containing the *N*-terminal group was examined for modification.

**MALDI-ToF analysis**. MALDI-ToF measurements were recorded using a Per-Septive Voyager STR MS with nitrogen laser (337 nm) and 20 kV accelerating voltage with a grid voltage of 90%. At least 300 laser shots covering the complete spot were accumulated for each spectrum. Sinapinic acid (20 mg mL$^{-1}$) in 50% acetonitrile with 0.4% TFA was used as matrix. Protein solution (0.5–1.0 mg mL$^{-1}$) was mixed with an equal volume of matrix and 2 µL of the resulting mixture was loaded onto a silver sterling plate target. Apomyoglobin, cytochrome C, and aldolase were used as calibration samples. Extent of modification was determined by subtracting the protein-initiator conjugates $m/z$ values from native protein $m/z$ and dividing by the molecular weight of the initiator (220.9 g mol$^{-1}$). Molecular weights of peptide fragments obtained in protein digests were determined after the solutions were purified by use of ZipTipC$_{18}$ microtips. Bradykinin fragment, angiotensin II (human) and insulin oxidized B chain (bovine) were used for calibration.

**Surface-initiated ATRP from immobilized protein-immobilized**. A suspension of DMA beads (1.5 mL) and carboxybetaine methacrylate (CBMA, 29 mg, 125 µmol, TCI America) in 100 mM sodium phosphate (4.5 mL, pH 8) in the synthesis vessel was sealed with a rubber septum and bubbled with nitrogen at room temperature for 30 min. 500 µL of deoxygenated catalyst solution (CuCl$_2$, sodium ascorbate, and 1, 1, 4, 7, 10, 10-hexamethyltriethylenetetramine (HMTETA, Supplementary Methods), similar conditions as in solution-based synthesis) was then added to the synthesis vessel under nitrogen. The mixture was sealed and shaken at room temperature for 1–2 h. The beads were washed with 100 mM sodium phosphate (pH 8, 5 mL × 5).

**Protein–pCBMA releasing from DMA beads**. Agarase solution (15 µL, 1 U µL$^{-1}$) was added to a suspension of obtained protein–pCBMA beads (1.5 mL) in 100 mM sodium phosphate (pH 6, 985 µL) and rotated at room temperature overnight (Supplementary Fig. 23). To release, 20 mM sodium citrate (3.5 mL, pH 3) was added and rotated at room temperature for 1 h. 100 mM sodium phosphate buffer (pH 5) was used for releasing AChE-pCBMA from beads due to irreversible inactivation of AChE at low pH (details in Supplementary Methods). The supernatant containing protein–pCBMA conjugates was separated from the beads by filtration or centrifugation. Protein concentration in the supernatant was determined by UV absorption protein or BCA protein assay.

**Native CT stability at pH 3**. Native CT (40 µM) was dissolved in 20 mM citrate buffer (pH 3) and incubated at 25 °C. At given time points, aliquots (10 µL) were removed and measured activity in 950 µL of 100 mM sodium phosphate buffer (pH 8) and 40 µL of suc-AAPF-pNA solution (10 mM in DMSO) at 25 °C. The residual activity was calculated as a ratio of initial rates of hydrolysis reaction at given incubation time over the initial activity at time zero, which monitored by recording the increase in absorption at 412 nm using an UV–VIS spectrometer.

**Solution-based synthesis of protein–pCBMA**. Solution-based synthesis of CT-pCBMA was carried out as described previously[15,17]. Briefly, a solution of CBMA (104 mg, 0.45 mmol) and protein-initiator (18–20 µmol of initiator) in 100 mM sodium phosphate (20 mL, pH 8) was sealed and bubbled with nitrogen gas in an ice bath for 30 min. One milliliter of deoxygenated catalyst solution (described above) was then added to the polymerization reactor under bubbling nitrogen. The mixture was sealed and stirred at room temperature for 2 h. The conjugate was isolated by dialysis with a 25 kDa molecular weight cutoff dialysis tube in deionized water in a refrigerator for 24 h and then lyophilized.

**Characterization of PARIS by FT-IR spectroscopy**. 100 µL of beads at each step of the PARIS synthesis were rinsed with deionized water (1 mL × 5), and then frozen and dried in vacuum. FT-IR spectra of each sample were obtained with an IR spectrometer using a KBr pellet.

**Characterization released PARIS CT-pCBMA**. Chemical structure of the released CT-pCBMA conjugate by PARIS was characterized by $^1$H NMR and FT-IR measurements. In the $^1$H NMR spectrum, the signals of polymer backbone chain were observed at 1.0–1.3 (3 H, methyl) and 2.1 ppm (2 H, ethylene). The signals of carboxybetaine side chain (Supplementary Fig. 11) can be observed. In the IR spectrum, the absorption of ester group on the polymer chain was observed at 1727 cm$^{-1}$, and the specific absorption of amide groups from CT were observed at 1643 ($\nu_{C=O}$: amide I) and 1550 cm$^{-1}$ ($\delta_{N-H}$: amide II).

**Cleavage of the grafted pCBMA from the conjugate**. The grafted pCBMA was cleaved by acidic hydrolysis from the conjugate. Protein–pCBMA conjugate (10–20 mg) and 6 N HCl aq. (4–5 mL) were placed in a hydrolysis tube. After three freeze–pump–thaw cycles, the hydrolysis was performed at 110 °C for 24 h in vacuum. The cleaved polymer was isolated by dialysis using a 1 kDa molecular weight cut off dialysis tube in deionized water and was then lyophilized. The molecular weight of the cleaved polymer was measured by GPC.

**Determination of conjugate hydrodynamic diameter**. DLS data were collected on a Malvern Zetasizer nano-ZS located in the Department of Chemistry, Carnegie Mellon University, Pittsburgh, PA. The hydrodynamic diameters of native protein and conjugate were measured three times (5 run to each measurement) in various buffers at room temperature. Reported values are number distribution intensities.

**Determination of conjugate Michaelis–Menten kinetics**. Suc-AAPF-pNA (0–125 µL of 9.60 mM in DMSO) was mixed with sodium phosphate buffer (865–950 µL of 100 mM, pH 8). Native CT or conjugates solution (10 µL, 3.9 µM of CT) was added to the substrate solution. The initial substrate hydrolysis rate was monitored by recording the increase in absorbance at 412 nm using an UV–VIS absorbance spectrometer with a temperature-controlled cell holder at 25 °C. Michaelis–Menten parameters were determined by nonlinear curve fitting of initial rate versus substrate concentration plots using Enzfitter software.

**Determination of enzyme thermostability**. Native CT and conjugates (1.5–2.0 mL, 3.9 µM of CT) were incubated in 100 mM sodium phosphate buffer (pH 8.0) at 50 °C. Aliquots (10 µL) were removed and activity was measured using suc-AAPF-pNA (40 µL of 9.6 mM in DMSO) in sodium phosphate buffer (950 µL of 100 mM, pH 8) by UV–VIS spectroscopy with a temperature-controlled cell holder. Residual activity was calculated as a ratio of hydrolysis rate at a given incubation time over the initial hydrolysis rate for each sample.

**Flow reactor CT immobilization on DMA beads**. DMA beads (1.5 mL) and 20 mM citrate (3.0 mL, pH 2) were sealed in the flow reactor with a rubber septum. Deionized water was introduced into the reactor by a peristaltic pump at room temperature at a flow rate of 1 mL min$^{-1}$ for 30 min. CT (2.0 mg mL$^{-1}$) in 100 mM sodium phosphate (pH 6 or 8) containing 0.05 v/v% Tween20 was introduced into the reactor by a peristaltic pump at a flow rate of 1 mL min$^{-1}$ for 30 min to bind the DMA beads. The beads were then washed with 100 mM sodium phosphate (pH 8) containing 0.05 v/v% Tween20 for 30 min.

**Flow reactor ATRP initiator modification on immobilized CT**. 200 mM NHS-Br in DMSO was introduced into the reactor by a syringe pump at a flow rate of 8 µL min$^{-1}$ for 30 min. The sample was then washed with 100 mM sodium phosphate (pH 8) for 30 min. An aliquot (20 µL of beads) was taken from the reactor for the BCA and fluorescamine assays.

**Flow reactor surface-initiated ATRP from immobilized CT**. A suspension of DMA beads (1.5 mL) and carboxybetaine methacrylate (CBMA, 29 mg, 125 µmol) in 100 mM sodium phosphate (4.5 mL, pH 8) was sealed in the synthesis vessel with a rubber septum and bubbled with nitrogen at room temperature for 30 min. Five hundred microliters of deoxygenated catalyst solution (CuCl$_2$, sodium ascorbate, and HMTETA) was then added to the synthesis vessel under nitrogen. The mixture was sealed and stirred at room temperature for 2 h followed by washing with 100 mM sodium phosphate (pH 8) at a flow rate 1 mL min$^{-1}$ for 30 min.

**Flow reactor CT-pCBMA release from DMA beads**. Agarase solution (15 µL, 1 U µL$^{-1}$) was added to a suspension of CT-pCBMA beads (1.5 mL) in 100 mM sodium phosphate (pH 6, 985 µL) and was rotated at room temperature overnight. To release, 20 mM sodium citrate (3.5 mL, pH 3) was added and stirred at room temperature for 1 h. The supernatant containing CT-pCBMA conjugates was separated from the beads by filtration. Protein concentration in the supernatant was determined by UV absorption and enzyme hydrolysis of *N*-Succinyl-Ala-Ala-Pro-Phe *p*-nitroanilide (suc-AAPF-pNA) using a standard curve with native CT.

**Data availability**. All relevant data is included in the manuscript, Supplementary Information, or may be available upon author's request.

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

## Acknowledgements

The authors acknowledge financial support provided by Carnegie Mellon University Center for Polymer-Based Protein Engineering.

## Author contributions

H.M. synthesized the solid and solution-based protein–polymer conjugates, synthesized model dyes, performed and analyzed the binding/release fluorescence studies, characterized the protein–polymer conjugates for hydrodynamic diameter and activity, performed thermostability testing, and designed the flow reactor. S.C. performed the trypsin digestion studies of the five proteins and performed and analyzed MALDI-ToF MS data. S.L.B helped to design experiments and drafted the manuscript. A.R and K.M supervised the project and provided guidance.

## Additional information

**Competing interests:** The authors declare no competing financial interests.

