## [Peer Review File · Nature Communications]

Reviewers' comments:

Reviewer #1 (Remarks to the Author):

The manuscript by Russell and Matyjaszewski describes a very interesting and potentially versatile way to construct protein/polymer hybrids using grafting-from technology on solid supports. The methods described are generally accessible even to the novice and should be of broad interest to those working with protein/polymer hybrid materials. Given the relative simplicity of the method, I would think that this methodology may entice researchers outside of the field to enter this scientific domain. While I found the idea clever and am very positive on the approach, I feel that quite a bit more should be done prior to publication in a journal of this tier.

- My major criticism is that the polymer conjugates are hardly characterized. The authors demonstrate activity of chymotrypsin conjugates and report a DLS value (without histogram data). This, however, tells us nothing about purity, dispersity of the conjugate (except in so far as DLS can do so), and dispersity of the polymers. At a minimum a PAGE gel, size exclusion chromatogram, and GPC of cleaved polymers should be performed. Kinetic experiments would be useful to determine polymerization kinetics versus those in solution. All of these data should be compared to modification in solution. Generally, I think the manuscript focused too much on protein immobilization and the flow reactor and not enough on the innovative polymer chemistry that is the most critical part of the study.
- The authors spend significant time verifying resin loading and initiator conjugation with a wide scope of proteins, however, the more interesting piece of the paper is the polymer chemistry, since the resin immobilization is known chemistry. In this case, only a single protein is used. The authors should demonstrate scope, as well, especially since the initiator materials were already made. The manuscript would be more authoritative if this experiment was completed along with the above characterization – especially for uricase and acetylcholinesterase as those proteins have functional utility.
- From an experimental standpoint, there are some potential issue that may make this difficult to translate to other labs. The cleavage is at pH 3. This is a very harsh environment for many enzymes to endure. Showing protein scope would help to alleviate this concern.
- Why is agarase added? Likely to disrupt some of the bead for full protein release? If there is an extra protein in the reaction mixture, then doesn't this scheme run into the same challenges as grafting-from approaches?
- Lastly, the flow reactor is a hot area of research for multi-step syntheses, but realistically protein/polymer hybrids are unlikely to reach the level of complexity of peptides or DNA. It is more likely that only a single polymer or a diblock will be necessary. So, is there really much utility to this?
- As a minor point, the introduction seems very specialized for a journal with very broad readership. For instance, PEGylation is never defined. It would be helpful to start with a bigger picture for the casual reader.
- MALDI should be used in conjunction with amino group measurement to determine initiator attachment.
- I would recommend to remove the speculation about hydrophobicity and reduced activity. It is just speculation and there are many examples of 'super-charging' proteins to make them more stable. The decrease in stability appeared to be pretty modest, anyhow.

Reviewer #2 (Remarks to the Author):

The authors propose a strategy for the general and reversible immobilization of proteins to solid supports, followed by subsequent functionalization and polymer growth steps by atRP. The end result is a protein-polymer conjugate and a new method to produce/purify them. Emphasis is placed on convenience, and the potential for automation.

The study emphasizes the simple nature of the purification of the final desired bioconjugate from the reaction mixture. This is generally a challenging process for typical coupling between polymer and proteins, because of the difficulty in separating conjugate from protein and excess polymer. However, conjugate purification is typically much easier when the polymers are grown from the protein, as is the case herein. In the present case, catalyst/monomer/etc could be simply removed by diafiltration, which arguably is much simpler than using reversible immobilization steps. Diafiltration is also amenable to automation. To me, the complexity and duration of solution synthesis of bioconjugates is over stated.

To my knowledge, this is the first example of solid-supported bioconjugate preparation by ATRP. However, the solid-phase pegylation of proteins for the purpose of simplifying purification has been around since ca. 2009. One recent example in *Biomaterials* 2014 Volume 35, Issue 19, Pages 5206-5215 exploits the histidine tag on recombinant proteins for immobilization of the protein, rather than the pH-sensitive chemistry used herein. Thus, statements such as "PARIS is a novel synthetic approach that, for the first time, allows solid-phase synthesis of protein-polymer conjugates" is not accurate and does not reflect available literature.

The authors go to great lengths to validate that their immobilization strategy is selective to the N-terminus. While I personally don't think it's necessary for the authors to demonstrate this (because I don't believe it's selective, and that this aspect is not detrimental to the strategy), I am not convinced with the author's data supports their claim. Firstly, the authors use model reagents to validate selective reaction based on amino acid pKa, but in reality proteins behave differently because of both steric hindrance and local environment. In fact, it is even possible to quantitatively predict pKa shifts of ionizable groups on proteins if the adjacent amino acids are known. Secondly, the authors attempt to use MALDI TOF MS in a quantitative manner to attest to selective immobilization at the N-terminus. MS cannot be used in this manner because when a peak is absent it is not possible to distinguish between the specific peptide being absent and ionization problems. Indeed, the MS spectra provided by the authors immediately shows how difficult it can be to acquire high quality and reliable MS data. Thirdly, the authors demonstrate that two proteins with unexposed N-termini bind to the support. This to me is the most striking piece of evidence for lack of selectivity to the N-terminus.

In many cases, polymers are conjugated to proteins to mask them from immune responses in the body. As such, from a conceptual point of view, the solid support may be 'blocking' a large surface of the protein from growing a polymer. Thus, the resulting conjugate may have large unprotected patches that might be detrimental to their use in that specific application. Indeed, while that stability of the conjugates towards stresses was examined, enzymatic degradation was not checked. Indeed, in the aforementioned *Biomaterials* paper, the His Tag was removed enzymatically after pegylation, indicating that this part of the protein is not protected by the polymer.

Figure 1 – error in the structure of the ATRP initiator..

Are the authors concerned about adverse oxidation of their protein due to use of DMSO in the initiator grafting step?

Reviewer #3 (Remarks to the Author):

This work describes the synthesis of protein-polymer conjugates on solid support. The authors call this approach "PARIS" for Protein-ATRP on Reversible Immobilization Supports. Grafting-to approaches with PEG are typically ill defined and require excess reagents. The authors utilize a

grafting-from approach with the protein on solid support. The work on protein modification on the N-terminus is interesting and informative and the paper is generally written to be easy to follow. Yet, I do not think this report as a whole is of high enough impact to warrant publication in Nature Communications. It is fairly specialized (engineering) and also has limitations. For example, I would think the protein-polymer conjugates can be purified fairly rapidly if only monomers, unreactive initiator and catalysts (i.e. small molecules) are being removed. Further, the solid support limits the amount that can be made and is thus may not be that relevant for any scale up – especially since the conjugation yield at pH 6 and released protein at this pH was low. Plus adding the protein to the resin and releasing it adds two additional steps and not all proteins may be amenable to attachment to resin. Although the decrease in protein inactivation with the enzyme added to solid support at pH 6 is much lower than in solution, the same effect could presumably be seen if the initiators were conjugated in a more selective fashion in solution phase. Further, there is a lack of characterization in some instances (no statistics, little small molecule and polymer structure characterization, etc.) In addition to these general comments, more specific comments are below.

1. Targeting the N-terminal amine is well known to have side reactions with other amino acids including lysines – it is one reason why PEG proteins are so disperse. How do the authors know this is not the case? It appears they only looked at 1c msms at particular fragments (N-terminal fragments) and not at other fragments.
2. References 19 and 20 are not the best representatives.
3. Line 100 –Supplementary Figure 1 is just a scheme and doesn't show the data suggested in the text.
4. Line 136 – describe sizes and structures
5. Line 154 – why were the proteins released at pH 3?
6. Extended Data Figure 2 is not very useful; maybe better if combined with Figure 3 - Instead it would be useful to see real data here for one or all of the proteins.
7. Data paragraph 161 on – for chymotrypsin and others, what other residues were modified on the resin? Presumably this can be determined by MS.
8. Figure 2 – the authors should comment on what happens to the protein that is not released by the gel. In some cases there is significant loss. It is mentioned on line 217 but not explained.
9. The authors should also comment on why in the case of Lysozyme and uricase there are more protein initiators at pH 6 than 10, which seems unusual.
10. Line 218 – I do not know how the authors can say that “did not increase polymer length” since the molecular weight is not directly measured but inferred by the overall size determined by DLS, which is not that accurate to begin with. The authors should show the molecular weight of their polymers (maybe by mass spec or other method). Also lines 220-221 – where is the dispersity data for the polymer that is mentioned? Without polymer and molecular weight dispersity data, it is hard to say that the pore size limit polymer size.
11. There are no statistics mentioned or shown for Figure 4 – this needed to demonstrate differences. Also the number of repeats should be provided in the caption and experimental. Statistics are required to demonstrate there are differences between the polymer protein samples that are mentioned in the text.
12. 335 and 396 experimentals. Direct characterization such as IR and maybe NMR and MS are not

shown for the beads.

13. Tabulation of the NMR peaks and integrations, J couplings, etc. from NMR spectra for various compounds should be provided in the experimental. It is hard for readers to gather this exact data from the NMRs provided in the supporting information. Also ^1H NMR alone is not enough for the compounds – really ^{13}C NMR, IR, HRMS or EA, etc. need to be provided.

Response to Reviewers' comments:

Reviewer #1 (Remarks to the Author):

The manuscript by Russell and Matyjaszewski describes a very interesting and potentially versatile way to construct protein/polymer hybrids using grafting-from technology on solid supports. The methods described are generally accessible even to the novice and should be of broad interest to those working with protein/polymer hybrid materials. Given the relative simplicity of the method, I would think that this methodology may entice researchers outside of the field to enter this scientific domain. While I found the idea clever and am very positive on the approach, I feel that quite a bit more should be done prior to publication in a journal of this tier.

My major criticism is that the polymer conjugates are hardly characterized. The authors demonstrate activity of chymotrypsin conjugates and report a DLS value (without histogram data). This, however, tells us nothing about purity, dispersity of the conjugate (except in so far as DLS can do so), and dispersity of the polymers. At a minimum a PAGE gel, size exclusion chromatogram, and GPC of cleaved polymers should be performed. Kinetic experiments would be useful to determine polymerization kinetics versus those in solution. All of these data should be compared to modification in solution. Generally, I think the manuscript focused too much on protein immobilization and the flow reactor and not enough on the innovative polymer chemistry that is the most critical part of the study.

These are very good points and are important to add to the manuscript. We performed many additional experiments in order to improve the manuscript. Further characterizations during the individual synthesis steps of CT-pCBMA conjugates have been added. Specifically, experiments were performed to characterize reversible protein immobilization to the beads at varying pH values, as a function of time. CT was immobilized at both pH 6 and pH 8 and subsequent release was performed by incubation at pH: 3, 4, 5, 6 over 60 min. The amount of released protein at the different pH values was measured using a bicinchoninic acid (BCA) assay. This new data shows that the amount of released protein increased with decreasing pH of the releasing buffer and ~90-100% of protein could be recovered through incubation at pH 3 for 60 min.

Characterization through FT-IR was also performed at each step of the conjugate synthesis. The presence or absence of anhydride, imine, ester, amide I, amide II, and COO⁻ peaks verified chemical conjugation over the entire synthesis from initial protein immobilization to conjugate release.

Further characterization of released PARIS synthesized CT-pCBMA conjugates was performed using ¹H-NMR, dynamic light scattering (DLS), and polymer cleavage from CT through acid hydrolysis followed by gel permeation chromatography (GPC). Peaks in the ¹H-NMR verify the chemical structure of the pCBMA polymer. DLS data for CT-pCBMA conjugates immobilized at pH 6 and 8 and subsequently released at pH 3 showed similar increases in hydrodynamic diameter (D_h), in comparison to native CT due to the addition of polymer. GPC of cleaved polymer from these conjugates showed M_n (PDI) values of 9,200 (1.27) and 8,200 (1.26) for conjugates initially immobilized at pH 6 and 8, respectively.

ATRP kinetics were also performed under various ATRP conditions and comparisons were made for PARIS and solution-based synthesized conjugates. We stress that this paper was not designed to optimize the activity of any of the protein polymer conjugates, but rather to prove that PARIS-synthesized and solution-synthesized protein polymer conjugates had similar activity and size. Kinetics were monitored through DLS measurements of the conjugates over one hour and through polymer cleavage and GPC at various time points. With a monomer concentration of 25 mM, DLS and GPC measurements showed that the polymerization was completed after 5 minutes of reaction for both solution and PARIS synthesized conjugates (i.e. no significant increase in conjugate D_h or cleaved polymer M_n over 5-60 min). PARIS-synthesized conjugates were similar in many ways to solution-synthesized conjugates, with the added benefit of lower dispersities. We also verified that synthesis of different size conjugates can be prepared using PARIS chemistry by performing the ATRP kinetics reaction at increasing monomer:initiator ratios (monomer concentration was varied from 12.5-100 mM for a fixed initiator concentration). PARIS conjugate D_h and cleaved polymer M_n were measured for each ATRP reaction over one hour and compared to solution-based synthesis. Hydrodynamic diameter increased as the monomer concentration increased. Cleaved polymer also showed increased molecular weight values with increasing monomer concentration. This showed that polymer length, and thus overall conjugate size, was easily tuned to a desired value by controlling monomer:initiator ratio in the ATRP reaction.

The authors spend significant time verifying resin loading and initiator conjugation with a wide scope of proteins, however, the more interesting piece of the paper is the polymer chemistry, since the resin immobilization is known chemistry. In this case, only a single protein is used. The authors should

demonstrate scope, as well, especially since the initiator materials were already made. The manuscript would be more authoritative if this experiment was completed along with the above characterization – especially for uricase and acetylcholinesterase as those proteins have functional utility.

Conjugate synthesis using acetylcholinesterase, uricase, avidin, and lysozyme were performed and characterized for size and activity. For all protein-polymer conjugates, we characterized the conjugates by MALDI-ToF MS for initiator modification, dynamic light scattering for size, gel permeation chromatography for cleaved polymer molecular weight and dispersity and enzymatic activity (all compared to solution-based syntheses of analogous conjugates). These proteins have varying sizes, number of lysine residues, quaternary structures, *N*-termini accessibilities, and sensitivities to acidic conditions. These new data showed that PARIS chemistry is applicable to a wide range of proteins.

From an experimental standpoint, there are some potential issue that may make this difficult to translate to other labs. The cleavage is at pH 3. This is a very harsh environment for many enzymes to endure. Showing protein scope would help to alleviate this concern.

This is definitely a valid point. To address this, stability assays were performed for native CT in pH 3 (20 mM citrate) buffer by measuring the residual activity over a 3-hour incubation period. Native CT maintained full activity indicating that the protein is not inactivated during the releasing step. Moreover, the releasing step is performed after polymer growth from the protein and we have shown in previous works (Biomacromolecules 2014, 15, 2817–2823; Biomacromolecules 2017, 18, 576–586) that conjugation of polymer stabilizes proteins at non-native pH and temperatures which would provide even more protection against the acidity of the releasing buffer.

As the reviewer implied, however, this stability to pH 3 may not be true for all enzymes, as was the case for acetylcholinesterase. Fortunately, conjugate release can also performed at pH 4, 5, or 6 (data now added to the manuscript). Thus, the releasing buffer can be tailored to an appropriate pH depending on the chosen protein. We also observed that release rate increased with decreasing pH and that ~90% of protein was released after 10 min at pH 3 which significantly decreased the amount of time the protein would be exposed to low pH.

In this revised and significantly altered paper we include the scope of PARIS-synthesized conjugates with a range of proteins that have similar activity to solution-protein ATRP enzymes after polymer conjugation and release.

Why is agarase added? Likely to disrupt some of the bead for full protein release? If there is an extra protein in the reaction mixture, then doesn't this scheme run into the same challenges as grafting-from approaches?

Yes. A very small amount of agarase was added to help achieve full protein release and improve yield, but release will still occur without the addition of agarase. If agarase is needed to increase yield (perhaps for larger conjugates), agarase and conjugates could be separated with size exclusion chromatography. Our main point, however, was just to demonstrate that agarase could enhance and accelerate release, but was not a necessity. We have reduced discussion of agarase in this revision.

Lastly, the flow reactor is a hot area of research for multi-step syntheses, but realistically protein/polymer hybrids are unlikely to reach the level of complexity of peptides or DNA. It is more likely that only a single polymer or a diblock will be necessary. So, is there really much utility to this?

The major advantage of PARIS currently is that it significantly reduces conjugate synthesis time in comparison to solution-based methods since days-worth of dialysis between each synthesis step are not needed and new data (as mentioned above) suggest that the synthesis time can be decreased even further by reducing time for the ATRP reaction and conjugate release steps. The addition of the flow reactor design would be appealing for industrial scale syntheses and automation. We also show that “grafting-from” is achievable using PARIS chemistry which offers many benefits over “grafting-to”, including higher and more controlled modification. PARIS chemistry would be highly beneficial for both high throughput screening and combinatorial chemistries. The activity and stability of conjugates after polymer conjugation with varying modification density, polymer length, polymer type, protein type, etc. is currently unpredictable. PARIS can be used as a high throughput screening method to synthesize a large volume of conjugates in a short amount of time to determine its overall effect on protein function for both therapeutic and synthetic synthesis applications. PARIS is also highly attractive for combinatorial chemistry. Conjugates with functional diblock polymers (e.g. pH or temperature responsive for sensing), new monomers, mixtures of monomers to create copolymers, etc. can be easily and quickly be created using PARIS. Thus, PARIS allows a library of conjugates to be synthesized and characterized.

As a minor point, the introduction seems very specialized for a journal with very broad readership. For

instance, PEGylation is never defined. It would be helpful to start with a bigger picture for the casual reader.

This point has been addressed in the revised manuscript.

MALDI should be used in conjunction with amino group measurement to determine initiator attachment.

This suggestion was appreciated and taken into account. We have performed fluorescamine assays in conjunction with MALDI in previous studies and we see similar results, however, we have additionally incorporated MALDI data of initiator modification for the 5 proteins studied. We are also exploring ESI which appears to be more informative than MALDI, especially since liquid chromatography can be coupled to ESI to first separate different species before mass detection which would allow a more quantitative analysis. Uricase did not give a signal on MALDI, however, due to its high charge. Fluorescamine data were presented for uricase only.

I would recommend to remove the speculation about hydrophobicity and reduced activity. It is just speculation and there are many examples of 'super-charging' proteins to make them more stable. The decrease in stability appeared to be pretty modest, anyhow.

We concur.

Reviewer #2 (Remarks to the Author):

The authors propose a strategy for the general and reversible immobilization of proteins to solid supports, followed by subsequent functionalization and polymer growth steps by atp. The end result is a protein-polymer conjugate and a new method to produce/purify them. Emphasis is placed on convenience, and the potential for automation.

The study emphasizes the simple nature of the purification of the final desired bioconjugate from the reaction mixture. This is generally a challenging process for typical coupling between polymer and proteins, because of the difficulty in separating conjugate from protein and excess polymer. However, conjugate purification is typically much easier when the polymers are grown from the protein, as is the case herein. In the present case, catalyst/monomer/etc could be simply removed by diafiltration, which arguably is much simpler than using reversible immobilization steps. Diafiltration is also amenable to automation. To me, the complexity and duration of solution synthesis of bioconjugates is over stated.

We thank the reviewer for these comments. Indeed, diafiltration can be used in solution-based conjugate synthesis instead of conventional dialysis and we have tried this method in the past. We found that we lose a lot of conjugate from the synthesis due to binding and clogging of the membrane. Other researchers in the field have also observed unpredictable binding of conjugates to diafiltration membranes. Purification using PARIS is much simpler, can be performed in a matter of seconds for each purification step, and is not dependent on protein size. This is particularly useful when performing automated flow reactor conjugate synthesis of large volumes.

To my knowledge, this is the first example of solid-supported bioconjugate preparation by ATRP. However, the solid-phase pegylation of proteins for the purpose of simplifying purification has been around since ca. 2009. One recent example in Biomaterials 2014 Volume 35, Issue 19, Pages 5206-5215 exploits the histidine tag on recombinant proteins for immobilization of the protein, rather than the pH-sensitive chemistry used herein. Thus, statements such as "PARIS is a novel synthetic approach that, for the first time, allows solid-phase synthesis of protein-polymer conjugates" is not accurate and does not reflect available literature.

This is a good point that we have addressed in the revision. Background on His tag has been added to the manuscript. The His tag approach is only available for specialized proteins that come with the His tag recombinantly added. PARIS does not have this requirement and is open for use by most proteins. We have carefully restated our belief in PARIS's novelty as follows: "PARIS is a novel synthetic approach that, for the first time, allows solid-phase synthesis of "grown from" protein-polymer conjugates".

The authors go to great lengths to validate that their immobilization strategy is selective to the N-terminus. While I personally don't think it's necessary for the authors to demonstrate this (because I don't believe it's selective, and that this aspect is not detrimental to the strategy), I am not convinced with the author's data supports their claim. Firstly, the authors use model reagents to validate selective reaction based on amino

acid pKa, but in reality proteins behave differently because of both steric hindrance and local environment. In fact, it is even possible to quantitatively predict pKa shifts of ionizable groups on proteins if the adjacent amino acids are known. Secondly, the authors attempt to use MALDI TOF MS in a quantitative manner to attest to selective immobilization at the N-terminus. MS cannot be used in this manner because when a peak is absent it is not possible to distinguish between the specific peptide being absent and ionization problems.

Indeed, the MS spectra provided by the authors immediately shows how difficult it can be to acquire high quality and reliable MS data. Thirdly, the authors demonstrate that two proteins with unexposed N-termini bind to the support. This to me is the most striking piece of evidence for lack of selectivity to the N-terminus.

In many cases, polymers are conjugated to proteins to mask them from immune responses in the body. As such, from a conceptual point of view, the solid support may be 'blocking' a large surface of the protein from growing a polymer. Thus, the resulting conjugate may have large unprotected patches that might be detrimental to their use in that specific application. Indeed, while that stability of the conjugates towards stresses was examined, enzymatic degradation was not checked. Indeed, in the aforementioned Biomaterials paper, the His Tag was removed enzymatically after pegylation, indicating that this part of the protein is not protected by the polymer.

We thank the reviewer for these comments and agree that they are all valid points and worthy of further explanation. Indeed, amino acid residues will react differently per their chemical and structural environment. We have previously published a study discussing how these factors can impact the reactivity of amino groups (lysine residues and N-terminus) depending on their microenvironment (*ACS Biomaterials Science and Engineering* **2017** 3 (9), 2086-2097). With the proteins evaluated in the present study, we have also considered the differences in pK_a , exposed surface areas for the amino groups and steric effects. Previous studies have shown that due to the unique microenvironment of the N-terminus, this amino group will typically display a lower pK_a when compared to lysine side chains. Moreover, several research groups have used this difference in ionization for site-specific modification at the N-terminus (*Pharmaceutical Research* **2003** 20 (5), 818-825; *Chemical Science* **2017** 8 (4): 2717–2722; *Nature Chemical Biology* **2017** 13, 697–705). The use of model reagents with amino groups with different pK_a values elucidates these differences by reactivity rates and how they can be used for specific binding onto the solid support. Of course, selective binding can only be attained when the N-terminus is exposed. For lysozyme and avidin the α -amino group is buried and hence unavailable to react with the solid support. In these cases, amino groups from lysine residues were found to react first with solid support, leading to unselective modification. In contrast, proteins where the N-terminus is exposed (chymotrypsin, uricase and acetylcholinesterase) selective modification of the α -amino groups to the solid support was observed. Regarding the use of MALDI-TOF MS as a technique to characterize, we agree with reviewer that MALDI can have some limitations in terms of ionization. In this study, we chose MALDI-TOF to analyze native and digested proteins due to its ease of use and speed of analysis. Furthermore, we have previously used this technique in our work (*Biomacromolecules* **2005** 6 (6), 3380-3387; *ACS Biomaterials Science and Engineering* **2017** 3 (9), 2086-2097) and found consistent results for the amino groups modified experimentally and what would be expected from their structural and chemical environment thus confirming our hypothesis of selective N-terminus binding. We are, however, currently exploring the use of ESI and LC-MS to further validate these results for our follow-up papers.

Having said all of that, we do concur with the reviewer that we were too definitive in interpreting selectivity data, so we have amended the manuscript to lessen the discussion on site-specific modification.

The point that the solid support could block potential modification sites that would lead to exposed protein for enzymatic degradation is something that should be taken into account, but is also application dependent. This is true for therapeutic applications of protein-polymer conjugates, but is less important for other applications such as enzyme stabilization for biocatalysis of synthetic chemicals. In the case of therapeutics, current protein-polymer conjugates use a "grafting-to" approach where attachment of the first polymer could also potentially sterically block an important site from being modified by a second polymer chain during synthesis which could then be exposed during use through polymer chain dynamic motions. Although the beads do block a small portion of sites from being modified, similar challenges occur with traditional approaches. PARIS also uses "grafting-from" which typically leads to higher grafting densities over the traditional "grafting-to" approach. Also, the attachment of proteins to the beads will be random (not all protein is going to bind in a "bad spot"), so you will end up with a mixture. As a side note, agarose-DMA beads can be purchased with varying pore sizes, which is something that we are exploring to optimize/control initiator modification and polymer growth.

Figure 1 – error in the structure of the ATRP initiator.

We looked at the structure again and did not see an error. If the reviewer could be more specific we will gladly address any oversight we made in the structure.

Are the authors concerned about adverse oxidation of their protein due to use of DMSO in the initiator grafting step?

That is a valid concern by the reviewer. The use of DMSO in the initiator grafting step was to prevent the hydrolysis of the NHS group. DMSO is frequently used to dissolve hydrophobic compounds, which are then diluted in aqueous solution as was in the initiator modification reaction. During the initiator reaction, a volume of 225 μ L is added to a total volume of 6 mL solution, corresponding to less than 4% DMSO concentration. Small volumes of DMSO (<10%) do not typically have an adverse effect on protein (*Biophys. Chem.* **2007** 131, 62–70).

Reviewer #3 (Remarks to the Author):

This work describes the synthesis of protein-polymer conjugates on solid support. The authors call this approach "PARIS" for Protein-ATRP on Reversible Immobilization Supports. Grafting-to approaches with PEG are typically ill defined and require excess reagents. The authors utilize a grafting-from approach with the protein on solid support. The work on protein modification on the N-terminus is interesting and informative and the paper is generally written to be easy to follow. Yet, I do not think this report as a whole is of high enough impact to warrant publication in Nature Communications. It is fairly specialized (engineering) and also has limitations. For example, I would think the protein-polymer conjugates can be purified fairly rapidly if only monomers, unreactive initiator and catalysts (i.e. small molecules) are being removed. Further, the solid support limits the amount that can be made and is thus may not be that relevant for any scale up – especially since the conjugation yield at pH 6 and released protein at this pH was low. Plus adding the protein to the resin and releasing it adds two additional steps and not all proteins may be amenable to attachment to resin. Although the decrease in protein inactivation with the enzyme added to solid support at pH 6 is much lower than in solution, the same effect could presumably be seen if the initiators were conjugated in a more selective fashion in solution phase. Further, there is a lack of characterization in some instances (no statistics, little small molecule and polymer structure characterization, etc.) In addition to these general comments, more specific comments are below.

We thank the reviewer for the comments. Naturally we do not concur with the view that the work is of specialized interest. The same could have been said about the underlying chemistry for peptide and nucleic synthesis on solid supports before devices were made to manage and orchestrate the chemistry. Also, the purification of conjugates from their precursors is a notoriously difficult and unpredictable process. The other reviewers have noted the general interest in this chemistry.

In terms of the amount of protein per ml of beads being an issue, we have clarified that PARIS is not designed to be an approach for scale up (although for peptide and nucleic synthesis, such chemistry is used at scale). Rather, we are excited because the automation of conjugate synthesis will allow scientists to generate large numbers of conjugates and thereby fully explore the biotic/abiotic synthetic space. We have also shown (new data added to manuscript) that 90-100% of protein is recovered from the immobilization and release steps, so scale-up and higher yields can be achieved. The only requirement for attachment of protein to the beads is that the proteins have a surface accessible group that reacts reversibly with chemistry on the bead. The decrease in yield at pH 6 vs pH 8 was only due to the site-specific N-terminal binding (less protein bound, thus less protein was released). Binding can also be slightly dependent on protein size, but pore size on the bead can be tuned for a protein of interest for higher loading. Loading optimization is not necessary for combinatorial and high throughput synthesis using PARIS.

Statistics were noted in the manuscript and extensive conjugate characterization has now been added. Further characterizations during individual synthesis steps of CT-pCBMA conjugates have been added. Specifically, experiments were performed to characterize reversible protein immobilization to the beads at varying pH values as a function of time. CT was immobilized at both pH 6 and pH 8 and subsequent release was performed by incubation at pH: 3, 4, 5, 6 over 60 min. The amount of released protein at the different pH values was measured using a bicinchoninic acid (BCA) assay. These new data show that the amount of released protein increases with decreasing pH of the releasing buffer and ~90-100% of protein can be recovered through incubation at pH 3 for 60 min.

Characterization through FT-IR was also performed at each step of the conjugate synthesis. The presence or absence of anhydride, imine, ester, amide I, amide II, and COO⁻ peaks verify chemical conjugation over the entire synthesis from initial protein immobilization to conjugate release.

Further characterization of released PARIS synthesized CT-pCBMA conjugates was performed using ^1H NMR, dynamic light scattering (DLS), and polymer cleavage from CT through acid hydrolysis followed by gel permeation chromatography (GPC). Peaks in the ^1H NMR verify the chemical structure of the pCBMA polymer. DLS data for CT-pCBMA conjugates immobilized at pH 6 and 8 and subsequently released at pH 3 showed similar increases in hydrodynamic diameter (D_h) in comparison to native CT due to the addition of polymer. GPC of cleaved polymer from these conjugates showed M_n (PDI) values of 9200 (1.27) and 8200 (1.26) for conjugates initially immobilized at pH 6 and 8, respectively.

ATRP kinetics were also performed for various ATRP conditions and comparisons were made for PARIS and solution-based synthesized conjugates. Kinetics were monitored through DLS measurements of the conjugates over the reaction time=60 min and through polymer cleavage and GPC at various time points over 60 min. With a monomer concentration of 25 mM, DLS and GPC measurements show that the polymerization was completed after 5 minutes of reaction for both solution and PARIS synthesized conjugates (i.e. no significant increase in conjugate D_h or cleaved polymer M_n over 5-60 min). This indicates that similar conjugates can be synthesized using PARIS as the traditional solution-based method and PARIS offers the additional benefit of lower dispersities. We also verified that synthesis of different size conjugates can be prepared using PARIS chemistry by performing the ATRP kinetics reaction at increasing monomer:initiator ratios (monomer concentration was varied from 12.5-100 mM for a fixed initiator concentration). PARIS conjugate D_h and cleaved polymer M_n were measured for each ATRP reaction over 60 min and compared to solution-based synthesis. Hydrodynamic diameter increased as the monomer concentration increased. Cleaved polymer also showed increasing molecular weight values with increasing monomer concentration. This shows that polymer length, and thus overall conjugate size, can be easily tuned to a desired value by controlling monomer:initiator ratio in the ATRP reaction.

1. Targeting the N-terminal amine is well known to have side reactions with other amino acids including lysines – it is one reason why PEG proteins are so disperse. How do the authors know this is not the case? It appears they only looked at lc msms at particular fragments (N-terminal fragments) and not at other fragments.

This is an important point and we have made sure to strength discussion of site of modification. Indeed, it is possible that side reactions with amino groups on lysine residues could react with the solid support leading to dispersity and lack of binding selectivity. However, due to the lower pK_a of the N-terminus (7.8 – 8.0) relative to the lysines ϵ -amino groups (10.5 – 12), we hypothesized a preferential binding of the N-terminus due to the higher reactivity of the α -amino group. This increased reactivity of the α -amino group was also confirmed with the results obtained from the pH dependence of binding studies with model compounds (Extended data Figure 2). These data showed that, at pH 6.0, GGcy3 (N-terminus) bound preferentially to the DMA beads while Cy5.5 amine (lysine) did not (see plots d and e). Additionally, for specific cases, such as lysozyme, where N-terminus selectivity was not observed, we analyzed further peptide fragments from the tryptic digest mixture and determined that lysines K116 and K33 were bound to the solid support. Thus, in the absence of an exposed α -amino group, the most reactive lysine residues will attach to the solid support. Previous literature for lysozyme also corroborated these results (*Biochimica et Biophysica Acta* **1989** 999, 1-6; *ACS Biomaterials Science and Engineering* **2017** 3 (9), 2086-2097) further validating our findings. We recently published a very detailed paper describing the techniques we used to determine how initiators react with proteins (*ACS Biomaterials Science and Engineering* **2017** 3 (9), 2086-2097).

2. References 19 and 20 are not the best representatives.

These references have been updated.

3. Line 100 –Supplementary Figure 1 is just a scheme and doesn't show the data suggested in the text.

This figure was intended to describe the synthesis scheme for the preparation of dialkyl maleic anhydride modified agarose beads. This has been changed in the manuscript to better represent this.

4. Line 136 – describe sizes and structures

Protein molecular weights and quaternary structure have been added.

5. Line 154 – why were the proteins released at pH 3?

Release at pH 3 was chosen because it provides the fastest release and highest protein recovery. Kinetics of release were performed (data added to manuscript) showing that in fact, full release of protein can be achieved at pH 3 after ~10 min of incubation (instead of the 1 hour incubation initially performed). Release can also be performed at pH 4, 5, or 6 with varying release rates for highly pH sensitive proteins. Acetylcholinesterase proved to be highly sensitive to pH, thus we performed the releasing step for AChE-

polymer at a higher pH.

6. Extended Data Figure 2 is not very useful; maybe better if combined with Figure 3 - Instead it would be useful to see real data here for one or all of the proteins.

Extended data figures have been revised in the resubmitted manuscript. The previous Extended Data Figure 2 has been taken out.

7. Data paragraph 161 on – for chymotrypsin and others, what other residues were modified on the resin? Presumably this can be determined by MS.

This is an interesting question by the reviewer. Yes, it is possible to determine what other residues were modified. However, given that the primary aim of this analysis was to determine selectivity of the N-terminus to the DMA beads, we focused on the analysis of that peptide fragment. When selectivity was not obtained, a more in-depth analysis was performed. For example, with lysozyme the lack of N-terminus binding led us to investigate which residues were in fact bound to the DMA beads. Trypsin digestion followed by MS analysis indicated that the most exposed, and therefore more reactive lysine residues (K116 and K33) were immobilized onto the solid support.

8. Figure 2 – the authors should comment on what happens to the protein that is not released by the gel. In some cases there is significant loss. It is mentioned on line 217 but not explained.

This is an interesting good point. We do observe some residual protein on the beads after release measured by IR, however, we can achieve almost full recovery of protein from release at pH 3. Recovery of conjugates however is always less than native protein. To combat this, either more agarose can be added to aid in disruption of the beads to promote release or, multiple release steps can potentially be performed in series. The focus of this manuscript was to present a new synthesis scheme for “grafted-from” conjugates, however, and further optimization of protein binding and release will be explored in the future by varying bead pore size.

9. The authors should also comment on why in the case of Lysozyme and uricase there are more protein initiators at pH 6 than 10, which seems unusual.

We thank the reviewer for this excellent suggestion. Indeed, for lysozyme and uricase the number of initiators coupled to the surface of the proteins at pH 6.0 was higher than at pH 8.0. For lysozyme, the unavailability of the N-terminus to react with the DMA beads led to protein immobilization occurring at non-selective lysine residues. While not included in the original manuscript, trypsin digestion studies suggested that immobilization of lysozyme on the solid support occurred at two different lysines, K116 and K33. This is consistent with previous studies that indicated these lysine residues as the most surface accessible, and therefore more available to react with the DMA beads. Multiple attachments of lysozyme can lead to different orientations of the protein on the solid support which can explain the differences observed in the number of initiators coupled on the surface of lysozyme. For uricase, N-terminus binding selectivity was observed and therefore the difference in the number of coupled initiators is unusual. Initiator modification was initially determined solely by fluorescamine assays. We have now re-examined the level of modification at each protein using more quantitative techniques (MALDI-ToF and/or ESI) and have incorporated the data.

10. Line 218 – I do not know how the authors can say that “did not increase polymer length” since the molecular weight is not directly measured but inferred by the overall size determined by DLS, which is not that accurate to begin with. The authors should show the molecular weight of their polymers (maybe by mass spec or other method). Also lines 220-221 – where is the dispersity data for the polymer that is mentioned? Without polymer and molecular weight dispersity data, it is hard to say that the pore size limit polymer size.

We thank the reviewer for highlighting these concerns. These data have been added to the revised manuscript. Briefly, conjugate characterizations were performed by dynamic light scattering and polymer cleavage through acid hydrolysis followed by GPC for molecular weight and dispersity. We also performed ATRP kinetics experiments with holding monomer concentration constant and performed reactions with increasing monomer concentrations. We observed that the ATRP reaction for pCBMA was completed within 5 minutes. We also showed that conjugate size (i.e. polymer length) could be varied by simply changing the monomer:initiator concentration. Overall, PARIS synthesized conjugates were highly similar to solution based conjugates.

11. There are no statistics mentioned or shown for Figure 4 – this needed to demonstrate differences. Also

the number of repeats should be provided in the caption and experimental. Statistics are required to demonstrate there are differences between the polymer protein samples that are mentioned in the text.

Error bars are represented on the stability figure, but are very small. An explanation of statistics has been added to the text. It is also worth mentioning that the stability of the conjugates is also similar to the stability of conjugates synthesized in the flow reactor design to help validate the data.

12. 335 and 396 experimentals. Direct characterization such as IR and maybe NMR and MS are not shown for the beads.

FT-IR data was taken at each step of the PARIS synthesis scheme to verify chemical conjugation (data added to extended data). NMR was also performed for conjugates to verify polymer chemistry. The structure of GGCy3 was verified in Supplemental Figures using NMR.

13. Tabulation of the NMR peaks and integrations, J couplings, etc. from NMR spectra for various compounds should be provided in the experimental. It is hard for readers to gather this exact data from the NMRs provided in the supporting information. Also ¹H NMR alone is not enough for the compounds – really ¹³C NMR, IR, HRMS or EA, etc. need to be provided.

These characterizations have been performed and updated in the manuscript.

REVIEWERS' COMMENTS:

Reviewer #1 (Remarks to the Author):

The authors did a more than admirable job addressing this reviewer's concerns. I think the revised manuscript is appropriate for publication and will be of broad readership.

Reviewer #2 (Remarks to the Author):

I am happy to see that peer review has improved the quality of this manuscript. I believe this manuscript will contribute to facilitating the synthesis of protein-polymer conjugates, especially within research groups that lack experience in chemistry. This will make these exciting materials more readily available for study, for many different applications. The authors have adequately addressed my comments and concerns. I feel that this manuscript is suitable for publication. BTW, I could not find the mistake in the structure of the ATRP initiator (Figure 1) mentioned in my previous appraisal. Either I was wrong or the version I was working with had a formatting error of some kind. If memory serves, there was an inconsistency in the structure of the NHS ...

Reviewer #3 (Remarks to the Author):

The authors have indeed added a lot of characterization data that was previously lacking. However, the claims are still overblown. For example, the way the abstract/intro is written indicates that PARIS is the way to make protein-polymer conjugates for a wide range of proteins. However, the pH 3 that is required for full recovery is still very low for most proteins. There is only ~45% recovery at a reasonable acidic pH. The models utilized, especially chymotrypsin are very stable. In addition, it takes 4 steps rather than 2 from the solution way the authors are preparing the conjugates (2 extra steps of protein immobilization and cleavage). For the other proteins investigated, the recovery is only 20%. Rather than just getting trapped in the pores, it is possible that the protein is degraded or unfolded during the process, showing limitation of scope.

Other comments:

In the newly added data – in Figure 2 is pH 4 and 5 may be swapped in caption.

Response to Reviewers' comments:

Reviewer #1 (Remarks to the Author):

The authors did a more than admirable job addressing this reviewer's concerns. I think the revised manuscript is appropriate for publication and will be of broad readership.

We would like to thank the reviewer again for the first round of remarks. It greatly improved the impact of the manuscript.

Reviewer #2 (Remarks to the Author):

I am happy to see that peer review has improved the quality of this manuscript. I believe this manuscript will contribute to facilitating the synthesis of protein-polymer conjugates, especially within research groups that lack experience in chemistry. This will make these exciting materials more readily available for study, for many different applications. The authors have adequately addressed my comments and concerns. I feel that this manuscript is suitable for publication. BTW, I could not find the mistake in the structure of the ATRP initiator (Figure 1) mentioned in my previous appraisal. Either I was wrong or the version I was working with had a formatting error of some kind. If memory serves, there was an inconsistency in the structure of the NHS ...

We would like to thank the reviewer for all of the remarks and suggestions throughout on ways to improve the manuscript. Implementing these changes increased the impact and scope of the paper.

Reviewer #3 (Remarks to the Author):

The authors have indeed added a lot of characterization data that was previously lacking. However, the claims are still overblown. For example, the way the abstract/intro is written indicates that PARIS is the way to make protein-polymer conjugates for a wide range of proteins. However, the pH 3 that is required for full recovery is still very low for most proteins. There is only ~45% recovery at a reasonable acidic pH. The models utilized, especially chymotrypsin are very stable. In addition, it takes 4 steps rather than 2 from the solution way the authors are preparing the conjugates (2 extra steps of protein immobilization and cleavage). For the other proteins investigated, the recovery is only 20%. Rather than just getting trapped in the pores, it is possible that the protein is degraded or unfolded during the process, showing limitation of scope.

Other comments:

In the newly added data – in Figure 2 is pH 4 and 5 may be swapped in caption

We thank the reviewer for their time and remarks. The suggestions that were made greatly improved the impact of the paper. We understand that there are limitations in this approach, as with any approach. Specifically, there is a trade-off between releasing pH and yield. If the release needs to be performed at a higher pH to maintain activity, such as 6.0, you will sacrifice the higher yield. We are confident that optimizing the conditions, however, (as mentioned around lines 271-276) will improve the recovery yield, but this was not the focus of this paper. The focus was showing that the chemistry was robust enough to synthesize a wide variety of protein-polymer conjugates that had similar activities as their solution-based counterparts. We show that for 2 proteins, lysozyme and uricase, the PARIS method actually had double the amount of activity in comparison to similar conjugates synthesized in solution. Also, the lower yield of the released conjugates in comparison to the studies with native protein led us to believe that it was the increase in size that was prohibiting full release from the bead pores and not due to protein degradation, although this could be a possibility that would need further investigation in future studies. We have also shown in past studies that the conjugation of polymers enhance the stability of proteins at pH as low as 1. Since the proteins are released at the lower pH only after polymer growth, we expect the polymer to help protect the protein from denaturation during release. Having said all of this, we have amended the manuscript in three places to try to ensure that the concerns of the reviewer are appropriately addressed.

The major steps for synthesizing conjugates in solution are:

1. Initiator modification
2. Dialysis for at least 1 day to remove unreacted initiator
3. ATRP
4. Dialysis for at least 1 day to remove monomer

The major steps for synthesizing conjugates by PARIS are:

1. Protein immobilization to beads
2. Initiator modification
3. ATRP
4. Conjugate cleavage from beads

Both methods have the same number of steps, but the biggest advantage of PARIS is the decrease in the amount of time it takes to end up with a purified conjugate (a few hours versus days). In PARIS, the washes between each step only take a couple of minutes. The other major advantage is that the set-up easily allows for automation, scale-up, and high-throughput screening.

Lastly, we also verified the accuracy of the legend in Figure 2.